# Supermagnetic Human Serum Albumin (HSA) Nanoparticles and PLGA-Based Doxorubicin Nanoformulation: A Duet for Selective Nanotherapy

**DOI:** 10.3390/ijms24010627

**Published:** 2022-12-30

**Authors:** Julia Malinovskaya, Rawan Salami, Marat Valikhov, Veronika Vadekhina, Aleksey Semyonkin, Alevtina Semkina, Maxim Abakumov, Yifat Harel, Esthy Levy, Tzuriel Levin, Rachel Persky, Vladimir Chekhonin, Jean-Paul Lellouche, Pavel Melnikov, Svetlana Gelperina

**Affiliations:** 1Drug Delivery Systems Laboratory, D. Mendeleev University of Chemical Technology of Russia, Miusskaya pl. 9, 125047 Moscow, Russia; 2Department of Chemistry, Faculty of Exact Sciences, Bar-Ilan University, Ramat Gan 5290002, Israel; 3Institute of Nanotechnology and Advanced Materials, Department of Chemistry, Faculty of Exact Sciences, Bar-Ilan University, Ramat Gan 5290002, Israel; 4Department of Neurobiology, V. Serbsky Federal Medical Research Centre of Psychiatry and Narcology of the Ministry of Health of the Russian Federation, Kropotkinskiy per. 23, 119034 Moscow, Russia; 5Department of Medical Nanobiotechnology, Pirogov Russian National Research Medical University, Ostrovityanova ul 1, 117997 Moscow, Russia

**Keywords:** hybrid maghemite/human serum albumin nanoparticles, PLGA nanoparticles, EPR effect, intravital microscopy (IVM), magnetic resonance imaging (MRI), 4T1 murine mammary carcinoma, 4T1 mScarlet cells

## Abstract

Predicting the ability of nanoparticles (NP) to access the tumor is key to the success of chemotherapy using nanotherapeutics. In the present study, the ability of the dual NP-based theranostic system to accumulate in the tumor was evaluated in vivo using intravital microscopy (IVM) and MRI. The system consisted of model therapeutic doxorubicin-loaded poly(lactide-co-glycolide) NP (Dox-PLGA NP) and novel hybrid Ce^3/4+^-doped maghemite NP encapsulated within the HSA matrix (hMNP) as a supermagnetic MRI contrasting agent. Both NP types had similar sizes of ~100 nm and negative surface potentials. The level of the hMNP and PLGA NP co-distribution in the same regions of interest (ROI, ~2500 µm^2^) was assessed by IVM in mice bearing the 4T1-mScarlet murine mammary carcinoma at different intervals between the NP injections. In all cases, both NP types penetrated into the same tumoral/peritumoral regions by neutrophil-assisted extravasation through vascular micro- and macroleakages. The maximum tumor contrasting in MRI scans was obtained 5 h after hMNP injection/1 h after PLGA NP injection; the co-distribution level at this time reached 78%. Together with high contrasting properties of the hMNP, these data indicate that the hMNP and PLGA NPs are suitable theranostic companions. Thus, analysis of the co-distribution level appears to be a useful tool for evaluation of the dual nanoparticle theranostics, whereas assessment of the leakage areas helps to reveal the tumors potentially responsive to nanotherapeutics.

## 1. Introduction

The concept of drug delivery to tumors by nanoparticles (NP) is largely based on the enhanced permeability and retention (EPR) effect that relies on the anatomical and pathophysiological differences between solid tumors and normal tissues. Thus, the EPR effect is a functional consequence of such tumor features as abnormally leaky blood vessels and impaired lymphatic drainage that are beneficial for its growth and also enhance the extravasation and retention of macromolecules and colloids in the tumor. First reported by Matsumura and Maeda in 1986 [1], the EPR effect initially served as a basis for the development of anticancer therapy using macromolecular drugs but later it was also applied to colloidal drug delivery systems based on nanocarriers, such as polymeric nanoparticles, liposomes, and micelles. [1,2]. However, the results of the EPR-based delivery as a strategy for enhancing the selectivity of the antitumor drugs are often disappointing. One possible reason is a considerable inter- and intra-individual heterogeneity of tumors. Indeed, while the EPR effect is undoubtedly an intrinsic pathophysiological feature of solid tumors, it may change over time during tumor development and varies not only between animal models and patients, but even among tumors and metastases within the same patient [3,4]. This heterogeneity of tumors and complexity of the EPR phenomenon might be responsible, at least in part, for the limited translation of colloidal drug delivery systems into clinics [5,6,7,8]. Moreover, the EPR mechanisms has been much disputed. Thus, the earlier studies described the nanocarriers’ entry into tumors as a result of their influx via the more or less permanent inter-endothelial gaps (or pores) in the tumor vessels, and it was therefore concluded that this influx depended on the cut-off size of these pores [9,10,11]. In contrast, the more recent results provide evidence that the permeability of the tumor vascular system is a dynamic phenomenon with transient leakages (or vents) and that the particle extravasation into the tumor and peritumoral area also depends on the nanoparticle–cell interactions [12,13,14]. Interestingly, as shown by Naumenko et al. [15], the appearance of bigger leakages (macroleakages) is associated with neutrophils that use these leakages as the route for extravascular escape and serve as Trojan horses for the liposomes trafficking them to the tumor site.

Numerous attempts have been made to improve the EPR-mediated tumor targeting by modulation of tumor vessel permeability using both pharmacological strategies (e.g., combination chemotherapy with nitric oxide donors or VEGF agonists and antagonists) and physical methods (e.g., ultrasound or hyperthermia) (reviewed in [2,4,16]). An alternative strategy that could help to improve the performance of nanotherapeutics in clinics involves a personalized approach to evaluation of the EPR effect manifestation in tumor(s) prior to chemotherapy, thus offering the possibility to select the patients potentially responsive to such treatment. As shown recently, this goal might be achieved by previous MRI (magnetic resonance imaging) of the tumors using diagnostic magnetic nanoparticles (MNPs) [17,18]. Thus, Miller at al. found colocalization of ferumoxytol (17–30-nm iron oxide nanoparticles coated with carbohydrate) with bigger 90-nm therapeutic PLGA nanoparticles in subcutaneous HT1080 human fibrosarcoma xenografts in nude mice.

Magnetic iron oxide nanoparticles have been studied extensively for numerous biomedical applications, such as, drug delivery system [19], magnetic targeting [20], and magnetic hyperthermia [21,22]. In particular, small and ultrasmall paramagnetic iron oxide nanoparticles (IONP) have a high potential for MRI diagnostics. Thus, several IONP-based contrasting agents have been approved by the FDA for the imaging of different pathologies, such as, for example, liver diseases (Feridex IV^®^ and Resovist^®^) and lymph node metastases (Sinerem^®^ and Combidex^®^). However, eventually, most of the IONP-based agents have been withdrawn from clinical practice with only few examples of current clinical application such as Combidex^®^ in the Netherlands [23]. The reasons for discontinuation are specific for each product but in general they may be outlined as insufficient safety that is not outweighed by clinical benefits [24]. Thus, ferumoxytol (Feraheme^®^, AMAG Pharmaceuticals, Waltham, MA, USA), an effective contrasting agent, that is currently approved as a medication for the treatment of iron deficiency but not for MRI, raises concern due to the risk of severe hypersensitivity reactions [25]. One way to increase the safety of the iron oxide-based contrasting agents is to increase their relaxivity and, therefore, decrease their dose necessary for tumor detection by MRI. As shown previously by Lellouche et al. [26], this possibility is offered by the water-compatible maghemite (γ-Fe_2_O_3_) NPs doped by Ce^3/4+^ cation arising from the magnetite oxidation step with ceric ammonium nitrate (CAN). The Ce^3/4+^-IONPs [26] are found to be effective for magnetic targeting [27]. They also proved to be useful as MRI imaging agents with an r2* relaxivity value of 189 mM^−1^s^−1^ (compared to ferumoxytol with r2* relaxivity value of 55.7 s^−1^mM ^−1^ in plasma). Further research demonstrated that exceptionally high r2* relaxivity could be achieved by incorporating these Ce^3/4+^-IONPs in a human serum albumin (HSA) matrix [28]. Such hybrid supermagnetic nanoparticles (hMNPs) exhibited an almost three-fold higher r2* relaxivity (482 mM^−^s^−1^) as compared to bare Ce^3/4+^-doped-maghemite nanoparticles, due to a clustering effect of IONP inside the HSA matrix, a phenomenon observed also by other authors [29].

Many researchers have focused on MNPs coating to improve their physicochemical and biological properties [30,31,32]. In this context, HSA, the most abundant plasma protein, appears to be a suitable candidate for MNP coating [33]. In general, using HSA as a drug carrier has been shown to improve the drug stability, pharmacokinetics, and biodistribution [34,35,36]. Recent work has demonstrated that single-core MNP with BSA effectively improves MNP stability and ensures a robust performance of MNPs as tracers in molecular imaging [37]. Others observed a greater stability for HSA–MNPs in saline and acidic media, preventing nanoparticle dissolution in extreme gastric conditions [38]. There are different strategies for the design of hybrid HSA–MNPs composites, such as creating a monolayer of HSA on single MNPs [37,38] and encapsulation of the MNPs within the HSA matrix before NPs are formed via desolvation and cross-linking [28] which is similar to the nanofabrication method described in this work. The approach used in the present study that is based on fabrication of magnetic nanoclusters comprised of small magnetic nanoparticles embedded in the HSA matrix produces both the biocompatible MNP surface and the enhanced MRI contrasting properties due to almost three-fold higher r2* relaxivity.

The objective of the present study was to evaluate the potential of the hMNP as a diagnostic companion for the model therapeutic poly(lactide-co-glycolide) nanoparticles loaded with an antitumor antibiotic doxorubicin (Dox-PLGA NP). This delivery system based on a biodegradable and biocompatible PLGA copolymer has previously exhibited a considerable antitumor effect against glioblastoma in rats [39] and confirmed its good tolerance in the clinical Phase I study in patients with solid tumors [40]. In the preclinical studies, the Dox-PLGA NPs also demonstrated a beneficial toxicological profile, including lower cardiotoxicity as compared to the free drug [41].

In this study, we analyzed the microdistribution of the fluorescently labeled hMNP and Dox-PLGA NPs in the vasculature and peritumoral area of both orthotopically and subcutaneously implanted 4T1-mScarlet murine mammary carcinoma using the intravital microscopy (IVM) and MRI. A special focus of the IVM imaging was placed on the role of vascular leakages in the nanoparticle microdistribution and the quantitative assessment of the prognostic value of hMNP. Accordingly, the 4T1 tumor served as an appropriate model for this study due to its average level of vascularization compared to other experimental tumors [42,43]. Additionally, the internalization of the hMNP and Dox-PLGA NP in the 4T1 cells was elucidated in vitro using confocal laser scanning microscopy (CLSM).

## 2. Results

### 2.1. Nanoparticle Preparation and Characterization

*Fluorescently labeled PLGA nanoparticles*. To achieve stable fluorescent labeling of the nanoparticles, PLGA was conjugated with the reactive dye derivatives Cyanine5 amine and Cyanine3 amine via the carboxylic end-group by the carbodiimide coupling method. The Cy5-labeled PLGA nanoparticles loaded with doxorubicin (Dox-PLGA-Cy5 NP) were prepared by the high-pressure *w/o/w* emulsification—solvent evaporation technique using a (1:1) mixture of the PLGA-Cy5 conjugate and non-modified polymer as described in [44]. The non-loaded PLGA NP labeled with Cyanine3 or Cyanine5 (PLGA-Cy3 NP and PLGA-Cy5 NP) were prepared similarly using the *o/w* emulsification—solvent evaporation technique. All PLGA nanoparticles had similar sizes of ~100 nm and a narrow size distribution (PDI below 0.2) (Table 1). The nanoparticle size distribution evaluated by intensity and volume (DLS data) is presented in Appendix A. The nanoparticles loaded with doxorubicin exhibited a slightly lower negative zeta potential as compared to the non-loaded nanoparticles (~−14.4 mV vs. ~−30 mV, respectively). As shown previously, the presence of the modified polymer did not influence the basic parameters of the PLGA NP [45].

*Fluorescently labeled hybrid supermagnetic nanoparticles.* The hybrid MNP (hMNP) were prepared by a two-step procedure. Firstly, the maghemite (γ-Fe_2_O_3_) NP were prepared, as described previously, by Massart’s basic co-precipitation of two types of Fe^2/3+^ salts followed by the ultrasound-assisted ceric ammonium nitrate (CAN) doping during the magnetite oxidation to its maghemite phase. This process yielded the magnetic NP doped with a (CeL_n_)^3/4+^cation/complex (CAN-MNP) [26,46]. In the next step, the hMNP were prepared by encapsulation of the CAN-MNP within the HSA matrix. For fluorescence imaging, the hMNP were labeled with either fluorescein or Cyanine3 (hMNP-FITC and hMNP-Cy3, respectively). For labeling, the HSA was conjugated with the commonly used reactive dye derivatives (fluorescein isothiocyanate or Cyanine3 NHS ester) prior to the hMNP formation (Figure 1). Thus, the functional protein groups enabled, on one hand, the formation of hMNP due to the HSA interaction with cerium ion/complexes of the CAN-MNP (primary amines of HSA attached coordinatively to cerium ion/complexes of IONPs) and, on the other hand, a stable fluorescent labeling of the hMNP via HSA conjugation of with the appropriate dyes.

In contrast to the positively charged CAN-MNP (+40–50 mV), both the hMNP-FITC and the hMNP-Cy3 had negative charges in the range of −(35–40) mV (Table 1), which is due to the negative HSA charge (isoelectric point at physiological pH is ~5). The hydrodynamic diameters of both labeled hMNP types were found to be approximately 140 nm with a PDI value of 0.1, indicating their narrow size distribution (Table 1).

As shown by scanning electron microscopy (SEM) and transmission electron microscopy (TEM), the PLGA NP had a spherical shape and were practically monodisperse (Figure 2d,e), which correlates with the above DLS data. The hMNP observed by TEM (Figure 2a–c) were non-spherical but had clear outlines with the encapsulated CAN-MNP seen as black dots. Figure 2a presents an accumulation of several particles, this image was obtained in the absence of uranyl acetate staining (grid preparation step) and therefore the CAN-MNP appear more contrast whereas the organic phase is blurred. The hMNP-FITC and hMNP-Cy3 observed by TEM was 30–45 nm (Figure 2b,c), which is considerably lower compared with their hydrodynamic diameter of >100 nm measured by DLS (Table 1; Appendix A). This phenomenon is most probably explained by the considerable hydrophilic shell acquired by the hMNP in aqueous media and the presence of a few aggregates that could also shift the light scattering intensity towards bigger size values. The size of the CAN-MNP encapsulated in the HSA matrix was about 7 nm.

### 2.2. Optical Properties of Nanoparticles

The purpose of this study required simultaneous visualization of different nanoparticle types in the biological environment using fluorescence microscopy methods. Therefore, several commonly used fluorescent dyes were employed to label various types of nanoparticles, namely FITC (λ_ex_ 491 nm, λ_em_ 516 nm), Cy3 (λ_ex_ 555 nm, λ_em_ 569 nm), and Cy5 (λ_ex_ 651 nm, λ_em_ 670 nm). This approach enabled reliable comparison of the following nanoparticle pairs: hMNP-Cy3 and PLGA-Cy5 NP, hMNP-FITC and PLGA-Cy5 NP, PLGA-Cy5 NP and PLGA-Cy3 NP.

As described above, all nanoparticles were covalently labeled with fluorescent dyes. In contrast to labeling by adsorption or encapsulation of a dye, covalent attachment prevents the dye from leaking out of nanoparticles thus minimizing the risk of misinterpretation of their biodistribution [45]. Optical properties were evaluated for all the nanoparticle types. The fluorescence measurements revealed that all nanoparticles exhibited similar values of brightness in the range from 3.3∙10^−5^ ℓ∙cm^−1^/mg to 4.2∙10^−5^ ℓ∙cm^−1^/mg (Appendix A), which proved sufficient for in vitro and in vivo visualization of the nanoparticles using the IVM and CLSM fluorescence microscopy methods.

### 2.3. Investigation of PLGA NP and hMNP Internalization into 4T1 Cells In Vitro

Internalization and intracellular trafficking of the fluorescently labeled PLGA-Cy5 NP and hMNP-FITC in the 4T1 murine mammary carcinoma cells was investigated using the fluorescent 4T1-mScarlet cell line generated by lentiviral transduction. Non-fluorescent 4T1 cells were used as a reference to confirm cell culture properties after transduction.The hMNP were added to the cells in the concentration of 200 μg/mL. As shown by the cytotoxicity assay (MTS test), this concentration did not significantly affect the 4T1 cell viability (Appendix A). Low in vitro cytotoxicity of the slightly different type of the hybrid HSA–MNP was also observed in our previous study [47]. Moreover, no significant increase in ROS (reactive oxygen species) generation in neutrophils was observed in response to the HSA–MNP which indicates that these NP are not likely to induce the oxidative burst in neutrophils. Low cytotoxicity of the PLGA NP was confirmed in a number of studies [48,49]. Accordingly, in our previous study the cytotoxicity of the PLGA-Dox NP in the U87 cells was comparable to that of free doxorubicin [44].

Both NP types preserved their initial size and polydispersity index within 24 h of incubation at 37 °C in PBS and serum-free RPMI cell medium.

#### 2.3.1. Internalization of Dox-PLGA-Cy5 Nanoparticles

At the first stage of this study, the internalization routes were investigated separately for each NP type—for the PLGA-Cy5 NP, Dox-PLGA-Cy5 NP, and hMNP. Efficient uptake of the Dox-PLGA-Cy5 NP by both 4T1 and 4T1-mScarlet cells was observed within 30–40 min of incubation. Due to the fluorescence properties of doxorubicin, its delivery to the cell nuclei could be visualized after 60 min of the Dox-PLGA-Cy5 NP incubation with the cells (Figure 3): the fluorescence intensity profile (Figure 3f) along the vector depicted by the orange arrow plotted on the image (Figure 3a–e) corresponds to heterochromatin of the nuclear DNA containing doxorubicin.

The intensity of doxorubicin fluorescence strongly depends on the biological environment, which can complicate interpretation of its imaging [50]; moreover, doxorubicin fluorescence spectra may interfere with the mScarlet fluorescence. Therefore, further experiments with 4T1mScarlet cells were performed using the non-loaded PLGA-Cy5 NP. The use of the model placebo PLGA-Cy5 NP also allowed for performing most of the experiments with all NP combinations, i.e., PLGA-Cy5 with both MNP-Cy3 and MNP-FITC, as well as PLGA-Cy3 in the control experiments. The degree of colocalization between the fluorophores was analyzed using the commonly used Manders’ overlap coefficient (MOC) or Pearson’s correlation coefficient (PCC) via pixel-by-pixel comparison of fluorescence intensities of two channels [51]. As shown by the analysis of confocal images (Appendix A), after 30 min of incubation with the 4T1 cells, the PLGA-Cy5 NP were only partly colocalized with lysosomes: Manders’ overlap coefficient (MOC) between lysosomes and PLGA-Cy5 was found to be below 0.5, whereas the colocalization of two fluorescent labels is considered reliable if the colocalization coefficients are above 0.5. Further, comparison of the MOC values between the two pairs—late endosomes/PLGA-Cy5 NP and lysosomes/PLGA-Cy5 NP—revealed that the MOC values for these pairs were 0.32 and 0.39, respectively, again indicating the absence of accurate colocalization between the PLGA NP and these organelles within 30 min of incubation (Appendix A).

#### 2.3.2. Internalization of PLGA-Cy5 NP and hMNP-Cy3 into 4T1 Cells upon Co-Incubation

At the next stage, internalization of the PLGA-Cy5 NP and hMNP-Cy3 into 4T1 cells upon their co-incubation was investigated in dynamics. As expected, different spectral characteristics of Cy3 and Cy5 enabled simultaneous visualization of the PLGA-Cy5 NP and hMNPCy3 and comparison of their intracellular distribution. The dynamics of internalization were investigated from the moment of the NP interaction with the cellular membrane (adhesion) to the moment of their detection in cellular organelles and cytoplasm. Interestingly, clear differences in the internalization rate and intracellular localization of the nanoparticles were observed: while the hMNP-Cy3 were visualized on the cell membranes and partly in cytoplasm within 20 min of incubation (Appendix A), the PLGA-Cy5 NP, on the contrary, were determined in cytoplasm mainly after 40 min of exposition (Figure 4). In addition, rare events of the hMNP-Cy3 colocalization with lysosomes were also detected (Appendix A). However, 30–40 min of incubation was found to be sufficient for the internalization of both NP types by the 4T1 cells.

Therefore, based on the above data, the colocalization between hMNP and PLGA NP was further assessed within 30–40 min of incubation (optimal time for intracellular visualization of both NP types). To investigate the nanoparticle localization within the cytoplasm, the Pearson correlation coefficient (PCC) between the following pairs of channels was determined: lysosomes—PLGA-Cy5 NP (Figure 5(1a–1c)), lysosomes—hMNP-Cy3 (Figure 5(2a–2c)) and also between two types of the nanoparticles—PLGA-Cy5 NP and hMNP-Cy3 (Figure 5(3a–3c)). The lysosomal tracker was chosen since the lysosomes are a well-established common eventual destination of the endocytosed cargo and also a marker of the clathrin-mediated endocytosis. In the histograms shown in Figure 5, the fluorescence intensities of the PLGA-Cy5 NP and lysosomes were plotted on the X- and Y-axis, respectively, providing a pixel-by-pixel comparison of the signal overlap. The PCC values were found to be 0.41, 0.33, and 0.48 for the lysosomes—PLGA-Cy5 NP (Figure 5(1c)), lysosomes—hMNP-Cy3 (Figure 5(2c)), and PLGA-Cy5—hMNP-Cy3 pairs (Figure 5(3c)), respectively. The PCC values of <0.5 were considered as weak colocalization or its absence [51].

Overall, these differences in the PLGA NP and hMNP internalization rates and intracellular localizations suggest that there might be no competition for the intracellular target between these nanoparticles upon their co-administration.

#### 2.3.3. Investigation of the PLGA-Cy5 NP and hMNP Internalization Pathway in 4T1 mScarlet Cells Using Endocytosis Inhibitors

The mechanism of the nanoparticle internalization was evaluated using commonly used inhibitors of the most important routes of NP internalization—the clathrin- and caveolin-dependent endocytosis pathways. Chlorpromazine, an amphiphilic cationic drug, is a commonly used inhibitor of clathrin-mediated endocytosis that anchors clathrin and inhibits clathrin-coated pit formation [52]. Methyl-β-cyclodextrin (MβCD) reversibly extracts cholesterol out of the plasma membrane and prevents caveolae formation thus inhibiting the caveolin-mediated (clathrin-independent) endocytosis that is dependent on the integrity of lipid rafts [53]. The activity of both inhibitors was evaluated by their incubation with the 4T1 cells prior to the NP addition using FITC-BSA as a reference molecule that is endocytosed predominantly by a caveolin-dependent route [54]. Additionally, dynasore, and amiloride were used in this experiment as the inhibitors of both clathrin- and caveolin-dependent endocytosis, and macropinocytosis, respectively. 

Effective inhibition of the PLGA-Cy5 NP uptake with MβCD (5 mM) is demonstrated in Figure 6. In contrast to control cells (Figure 6e,g), in the presence of MβCD, the PLGA-Cy5 NP were localized selectively on the cell membranes with only a few of them observed in the cell cytoplasm (Figure 6f,h), indicating that internalization of these NP is partly inhibited by MβCD. At the same time, nystatin and amiloride also affected the cellular uptake of these nanoparticles, but their effects were considerably less significant (not shown). Chlorpromazine did not affect the cellular uptake of the PLGA-Cy5 NP: the fluorescence intensity in the cell cytoplasm was comparable to that of the control. Therefore, caveolin-mediated endocytosis seems to play a major role in the PLGA NP internalization.

In addition, a significant decrease in the hMNP uptake by the 4T1 cells after cell preincubation with MβCD was also observed (Figure 6d). Similar to the PLGA-Cy5 NP, the hMNP were mainly localized on cell membranes. Similar results were obtained after the cell incubation with FITC-BSA (Figure 6b). However, while MβCD was found to be the most effective internalization inhibitor for both PLGA NP and hMNP, the accumulation patterns of these nanoparticles were different. Thus, in the cells treated with MβCD, the hMNP uptake was almost blocked; in contrast, the PLGA-Cy5 NP internalization was decreased but these NP were still detected in the cell cytoplasm.

Thus, the caveolin-mediated endocytosis appears to play an important role in the internalization of both hMNP and PLGA NP. These observations correlate with the results of other studies showing that the nanoparticles are internalized into the 4T1 cells via multiple mechanisms with the caveolin-dependent endocytosis pathway being one of the main internalization routes [55,56].

### 2.4. Investigation of hMNP In Vivo Distribution in the 4T1 Tumor Using MRI

Prior to the in vivo evaluation, the hemocompatibility of the hMNP was tested using routine colorimetric tests as described previously [57]. The hMNP did not affect the integrity of red blood cells in the concentrations up to 100 µg/mL, which is considerably higher than the expected blood concentrations of the hMNP in the MRI study (Appendix A).

In vitro evaluation of the hMNP magnetic properties by MRI revealed their considerably higher T2 relaxivity as compared to the once clinically approved contrasting agents based on the superparamagnetic IONP embedded in the carbohydrate matrix. Thus, the relaxivity coefficient r2 (1/T2) determined for the hMNP-FITC in water was found to be 473 mM^−1^s^−1^ (Appendix A), which is considerably higher than the r2 values reported for Resovist [58] and Ferumoxtran-10 [59] or ferumoxytol [60] (151 mM^−1^s, 60 mM^−1^s^−1^, and 89 mM^−1^s^−^1, respectively).

Accumulation of the hMNP at the tumor site and their contrasting potential was assessed in mice bearing subcutaneously and orthotopically implanted 4T1 tumors on day 10 after tumor inoculation. The T2-weighted images were captured 1 h, 5 h, and 24 h after the administration; maximum tumor contrasting with well-defined vasculature was observed at 5 h in both subcutaneously and orthotopically implanted tumors (representative images are shown in Figure 7 and Appendix A). By 24 h, the hMNP signals tended to shift to the tumor periphery. Apart from the tumors, the considerable hMNP accumulation was also observed in the spleen and the liver.

High contrasting properties of the hMNP were clearly visualized by their comparison with the Absolute Mag™ Carboxyl Dextran Magnetic nanoparticles (CD Bioparticles, USA, particle size 100 nm) that are structurally similar to conventional iron oxide-based contrasting agents (Figure 7). Both agents were injected in the same dose (3.5 mg/kg as Fe); therefore, the visible advantage of the hMNP is most probably explained by its considerably higher r2 relaxivity as compared to the Absolute Mag™ nanoparticles (473 mM^−1^s^−1^ versus 173 mM^−1^s^−1^, respectively).

The improved contrasting of the tumor obtained using the nanoparticle-based MRI agent suggests that the 4T1 tumor is an EPR-positive tumor model [61]. It is noteworthy that the nanoparticle accumulation patterns in tumors differed even among the animals of the same group (not shown), which confirms the heterogeneity of the EPR effect in the 4T1 tumors. Based on the obtained results, in further experiments the time point of 5 h was chosen as the optimal time to assess the hMNP accumulation in the tumor.

### 2.5. Investigation of the Nanoparticle Distribution in Mice Bearing Subcutaneous 4T1 Tumor Using Intravital Fluorescence Microscopy

#### 2.5.1. Evaluation of Blood Circulation Half-Lives

The blood circulation half-lives of the PLGA-Cy5 NP and hMNP-Cy3 were assessed by their fluorescence intensity dynamics within the ROI (regions of interest) selected in vessels of different diameters in the course of 40 min post injection. Both NP types were quickly disappearing from the blood; however, the circulation half-life of the hMNP-Cy3 was twice as long compared with the PLGA-Cy5 NP: 29 min and 55 min, respectively (Appendix A). Considering the similarity of the NP in terms of their sizes and zeta potentials (Table 1), this difference is most probably explained by a more hydrophilic, proteinaceous surface of the hMNP as compared to the PLGA NP consisting of a hydrophobic polymer, which makes the hMNP less prone to the uptake by the macrophages of the mononuclear phagocyte system and, consequently, prolongs their circulation in the bloodstream. Similarly, as shown by Bazile et al., coating of the polylactide NP with HSA considerably enhanced their plasma concentrations as compared with the uncoated NP which was attributed to the surface hydrophilization by the protein [62].

#### 2.5.2. Investigation of the Nanoparticle Microdistribution in Tumor Vasculature and Peritumoral Area in 4T1-mScarlet Tumor-Bearing Mice upon Simultaneous Administration

The PLGA-Cy5 NP and hMNP-FITC distribution and extravasation in the tumor vasculature was investigated from the moment of the tail vein injection (simultaneous NP administration) up to 45 min of imaging (Figure 8). In the first stage of distribution (min 1), only the vessel walls stained with CD-31-PE antibodies were observed (Figure 8(1a–1c)). Then both the PLGA-Cy5 NP and the hMNP-FITC were visualized within the vasculature (Figure 8(2a–2c)). The nanoparticle extravasation (oversaturated regions) via the vascular leakages was observed starting from 10–15 min after the administration and becoming more pronounced within the following 15 min. As seen in Figure 8(4), the fluorescent intensity profiles measured along the vector (white arrow) indicated high similarity in the microdistribution patterns of the PLGA NP and hMNP. Both nanoparticle types could be detected on the vessel walls and in the perivascular area. The dynamics of the nanoparticle extravasation via local microleakages to the perivascular area are demonstrated in Appendix A. Remarkably, 6 h post injection, the fluorescent signals in the vascular system from both NP types disappeared; only the signals from the stable areas of the NP extravasation were well determined for both the PLGA NP and the hMNP. The leakage areas and colocalization between the NP were observed even 24 h after the administration, although the fluorescence intensity decreased significantly by this time.

#### 2.5.3. Investigation of hMNP and PLGA NP Microdistribution in Tumor Vasculature and Peritumoral Area in 4T1-mScarlet Tumor-Bearing Mice upon Different Treatment Regimens

While the high colocalization level determined for the hMNP and the PLGA NP upon simultaneous administration appeared to be a promising result, this regimen has little or no value in terms of its practical application. Indeed, if the objective of the suggested strategy is to predict the patient susceptibility to treatment, then the diagnostic NP must be injected prior to their therapeutic companion. Moreover, simultaneous administration of the nanoparticles could alter their clearance profile due to their competition for the binding sites on the vessel walls as well as the immune cells. Therefore, colocalization of the hMNP and the PLGA NP was evaluated using other administration regimens, when the PLGA NP were injected 3, 5 or 24 h after the initial hMNP dose. The administration schemes were chosen with the consideration of the preliminary results showing the different times of the maximum hMNP accumulation in the tumor depending on the detection method. Thus, the maximum tumor contrasting was determined by MRI 5 h after the i.v. injection, whereas the maximum hMNP fluorescence intensity was observed by IVM 3 h after the i.v. injection.

Therefore, to address this task, the following administration schemes were used: (1) IVM evaluation performed 4 h after hMNP injection and 1 h after the PLGA-Cy5 NP injection (5 h/1 h); (2) IVM evaluation performed 2 h after hMNP injection and 1 h after PLGA-Cy5 NP injection (3 h/1 h); (3) IVM evaluation performed 23 h after hMNP injection and 1 h after PLGA-Cy5 injection (24 h/1 h). Depending on the fluorophores used in the experiments (fluorescently labeled antibodies, fluorescent or non-fluorescent 4T1 cells), the hMNP were labeled with either FITC or Cy3.

Consistent with the previous data on liposome microdistribution [15], extravasation of the hMNP and PLGA NPs in the peritumoral area occurred via the macro- and microleakages (representative images are shown in Figure 9). The dynamics of the macro- and microleakages formation are demonstrated in Appendix A. The differences in the NP accumulation patterns were observed for both 5 h/1 h and 3 h/1 h administration schemes. As seen in Figure 9(2a), the areas of the NP colocalization were rare with only few regions of the overlap between the hMNP-FITC and the PLGA-Cy5 NP fluorescent signals (Figure 9(2a)). Surprisingly, in contrast to the results obtained in the case of simultaneous administration, the accumulation areas of the hMNP-FITC and the PLGA-Cy5 NP administered with an interval between the doses seemed to be mutually exclusive (Figure 9(1a–1c)).

The similar tendency of the NP distribution was observed when the hMNP were administered 5 h prior to the PLGA NP: the large areas of the hMNP accumulation did not overlap with the leakage areas of the subsequently injected PLGA NP (Appendix A). Despite the fact that the hMNP accumulation in the tumor was still observed 24 h after i.v administration by MRI, only trace amounts of the hMNP were detected by IVM at this time (scheme 24 h/1 h). The signal from the hMNP was preserved only in the leakage areas (Appendix A). The hMNP-FITC and PLGA-Cy5 fluorescent signals were observed in the similar areas; however, they did not colocalize at the cellular level.

To exclude the possible influence of the differences in the NP surfaces (HSA coating vs. PLGA) on their distribution behavior upon subsequent administration, the PLGA NP labeled with different fluorophores (PLGA-Cy5 NP and PLGA-Cy3 NP) were administered as the first and the second dose (both 50 mg/kg). The 3 h/1 h administration scheme was chosen for this control experiment based on the maximum fluorescence intensity observed for both NP types within these time intervals. Due to different wavelengths of the dyes’ excitation/emission, the PLGA-Cy5 NP and PLGA-Cy3 NP were readily distinguishable by IVM. Interestingly, their distribution pattern was similar to the one observed previously for the hMNP-FITC and PLGA NPs administered at the same regimen (Appendix A).

These findings indicate the important role of macroleakages in the NP extravasation in the tumor and its environment. Indeed, as shown by Naumenko et al. [63] in contrast to relatively stable vascular microleakages the appearance of macroleakages is a dynamic process. Therefore, the observed difference between the microdistribution patterns of the NP doses administered at the intervals are most probably explained by their extravasation via the macroleakages that are not stable in space and time. Therefore, one can probably expect a low level of colocalization between the NP administered at intervals.

Interestingly, the hMNP appeared to accumulate in the pericyte-like cells wrapping around the endothelial cells (Appendix A), whereas the PLGA-Cy5 NP were only determined in the regions surrounding these cells, but never accumulated inside them. This phenomenon was apparently the key difference in the accumulation pattern of the PLGA NP and hMNP. 

To confirm the role of neutrophils in NP extravasation the fluorescence intensity of Ly6G-positive neutrophils in leakage areas was evaluated (detailed investigation of PLGA NP interaction with immune cells is described in Kovshova et al. («How is Doxorubicin Delivered to the Tumor? Drug Delivery Lifecycle»; manuscript in preparation). Fluorescence intensity profile (Figure 10) was measured only in the ROI (depicted by blue line) demonstrating the dynamics of NP extravasation through macroleakage (shape of ROI corresponds to that of the leakage area).

The correspondence of the fluorescent intensity peaks of the Ly6G-positive neutrophils, hMNP, and PLGA NPs was observed starting from 25 min after NP injection (indicated by white dashed line on the fluorescent intensity profile measured in dynamics (0–45 min after NP injection)) (Figure 10f). Thus, consistent with the previous data obtained for liposomes [15], the leakages of both NP types (PLGA and hMNP) were found to be neutrophil-associated, confirming the role of neutrophils in NP delivery to the tumor site.

The first step of the IVM study was performed for both non-loaded PLGA-Cy5 NP and the doxorubicin-loaded Dox-PLGA-Cy5. The distribution of the Dox-PLGA-Cy5 NP within the vasculature and leakage formation is demonstrated in Figure 11. The areas of the released doxorubicin accumulation in tumor tissue are seen in cyan (Figure 11a,c,e). Both NP types demonstrated similar patterns of the biodistribution and leakage formation (based on PLGA-Cy5 fluorescence) thus suggesting that the PLGA-Cy5 NP is a suitable model that could replace the drug-loaded Dox-PLGA-Cy5 NP in the IVM experiments. This finding was important, since doxorubicin with its broad fluorescence spectra could interfere with several additional fluorophores used in this study (i.e., fluorescently-labeled antibodies used to trace the vascular wall and immune cells). The possibility to use the PLGA-Cy5 NP as model NP helped to develop a more convenient and reliable experimental design. Thus, the experiments with different administration regimens were performed using the non-loaded PLGA-Cy5 NP.

#### 2.5.4. Quantitative Evaluation of the Prognostic Potential of the Theranostic Pair

However, despite the obvious absence of colocalization at the cellular level the ROI (tissue regions of ~2500 μm^2^) randomly selected in the areas of the NP accumulation appeared to contain both hMNP and PLGA NP fluorescent signals, which is indicative of the hMNP predictive potential. In contrast to colocalization that is generally understood as the overlap between distinctive labels in images, the above distribution pattern may be defined as “co-distribution”. Along with the colocalization level, the level of co-distribution can serve as a parameter for evaluation of the theranostic pair suitability.

To further prove this assumption, the leakage areas of both NP types within the ROIs were quantified. The nanoparticle microdistribution, leakage areas, and numbers were analyzed via the generation of binary masks in the ROIs selected for each frame (Appendix A). The ROIs containing the representative blood vessels (CD31) were generated randomly and characterized by the same size. Quantitative evaluation of the nanoparticle microdistribution was performed for two administration regimens. As mentioned above, the 5 h/1 h regimen (5 h after hMNP injection/1 h after PLGA-Cy5 injection) was chosen based on the MRI data of the maximal hMNP accumulation at the tumor site, whereas the 3 h/1 h regimen (3 h after hMNP injection/1 h after PLGA-Cy5) was found to be the optimal scheme regarding the hMNP brightness. A linear regression analysis revealed the positive correlation between the hMNP and PLGA NP distribution patterns for the 5 h/1 h regimen (Figure 12). Since the highest colocalization level between the PLGA NP and the hMNP was observed upon their simultaneous administration, this administration scheme was used as a positive control.

Accordingly, approximately 89% and 78% of both PLGA NP and hMNP were detected within the same ROI for 3h/1h and 5h/1h administration regimens, respectively, whereas for the simultaneously administered NP, this parameter reached 96%. These data confirm a high incidence of the NP co-distribution to the same regions for different administration regimens.

Overall, these findings demonstrate that the hMNP could predict the distribution and accumulation pattern of the PLGA NP, when the hMNPs were administered 5 h prior to the PLGA NP. This time, the interval also correlated with the period of maximum hMNP accumulation observed by MRI [42,43].

Importantly, IVM allows for evaluating not only the NP accumulation at the tumor site but also the process of their translocation across the vascular wall (large endothelial fenestrations) in the tumoral and peritumoral areas. Thus, it was possible to investigate the formation of the NP extravasation (leakage) areas, as well as to evaluate the leakage sizes and numbers. Such monitoring of the nanoparticles’ accumulation via determination of the leakage areas could presumably give a more comprehensive picture of the nanoparticle microdistribution pattern rather than the general comparison of the EPR effect for different nanoformulations. These observations also confirm that the 4T1 tumor is a suitable model for evaluation of the nanoparticle extravasation and accumulation in the tumor and peritumoral area.

## 3. Discussion

Over the last decades, several platforms for theranostics have been suggested for cancer treatment [42,43]. A popular approach involves the combining of the drug and the imaging agent (or even a combination of two imaging agents for multimodal imaging) in one targeted nanoformulation aiming to provide simultaneous drug delivery to the tumor and precise diagnostics [64,65,66]. The disadvantage of such theranostics is their technological complexity, leading to challenges in the technology scale up and transfer to industry as well as clinical translation of the product [67].

Another possible approach is to use a dual theranostic platform when the diagnostic nanoparticles are administered prior to the drug-loaded nanoparticles, which allows for selecting patients potentially responsive to the chemotherapy using nanotherapeutics prior to treatment. Furthermore, this technology is simpler and more cost-effective than multimodal nanosystems. As mentioned above, the feasibility of this approach was demonstrated by Miller et al. who used ferumoxytol (30-nm magnetic nanoparticles) to predict the tumor uptake and efficacy of the paclitaxel-loaded PEG-PLGA NP [17].

In the present study, a further insight into this approach is gained through the use of a theranostic pair consisting of the hybrid supermagnetic iron oxide-based hMNP [28] and the PLGA NPs as the diagnostic and therapeutic agents, respectively. Due to the enhanced magnetic properties of the hMNP, the dose required for reliable tumor MRI could be reduced as compared to conventional IONP-based contrasting agents, which, together with the biocompatible HSA surface, suggests that the hMNP are a potentially suitable and safe contrasting agent.

Evaluation of the fluorescently labeled diagnostic hMNP and therapeutic PLGA NP using the IVM technique enabled a comprehensive analysis of their spatial distribution and accumulation in the 4T1 tumor and peritumoral area. The 4T1 tumor model used in the present study appears to be suitable for the preliminary evaluation of the theranostic pair potential.

Special attention was given to investigation of the NP microdistribution using clinically relevant regimens when the therapeutic PLGA NP were administered within different time intervals after the hMNP. The intervals were chosen based on the preliminary MRI data regarding the hMNP accumulation in the tumor. Whether administered simultaneously or with an interval between injections, both NP types appeared to extravasate into the tumor or peritumoral area via micro- and macroleakages. This type of NP extravasation via the dynamic EPR effect (formation of transient vascular bursts or leakages) was previously described in a number of studies [13,16,68]. In particular, the distinct characteristics of these phenomena were described by Naumenko et al. [15] for the fluorescently labeled magnetic PEGylated liposomes: the microleakages were observed as frequent events of local liposome deposition to a perivascular area appearing within the few minutes after injection, whereas the macroleakages were more rare extravasation events covering vast interstitial area (hundreds of microns). In the present study, this type of extravasation is shown for the first time for a very different type of nanoparticles—solid, non-elastic and non-PEGylated, such as the PLGA NP and hMNP. Thus, the short-circulating PLGA NP with only a 29-min circulation half-life were able to effectively extravasate into the tumor within the first minutes after the administration.

Similarly, to the microdistribution pattern observed for PEGylated liposomes in the same tumor model [15], the NP extravasation areas were more frequently detected in the peritumoral area rather than in the tumor core which is obviously due to the higher vascularization of this region; however, it is the peritumoral area where the most active tumor proliferation process occurs, requiring more selective drug delivery to this region [16].

The IVM data obtained in the present study demonstrated that a high level of colocalization (overlap of the fluorescent signals between the diagnostic hMNP and the therapeutic PLGA NP in leakage areas) was only observed upon simultaneous NP administration. However, administration of the PLGA NP 2 h, 4 h or 23 h after the hMNP injection showed that on the cellular level the NP accumulation areas were almost mutually exclusive. Thus, to further investigate this phenomenon and evaluate the prognostic potential of the hMNP and possible limitations of this approach, we applied the method of regional assessment of the NP localization in tumor tissue: the leakage areas of both NPs were quantified within the randomly selected ROIs. Interestingly, despite the absence of NP colocalization within the cells, fluorescent signals of both the hMNP and PLGA NP were determined in most of the ROIs, demonstrating significant correlation in the NP microdistribution and accumulation patterns.

The significant colocalization between the PLGA NP and the hMNP within the first hour after simultaneous administration appears to be an interesting phenomenon. In general, the fate of the nanoparticles upon their entry in the bloodstream depends on their size, charge, shape, and surface [60,61,62,69]. In particular, the surface properties define the composition of the protein corona acquired by the nanoparticles from the blood. This corona has a direct impact on the nanoparticle interaction with different cells and the host’s immune system. Therefore, the surface properties and, especially, its hydrophobicity play an important role in the NP pharmacokinetics, biodistribution, and accumulation in tumors. Indeed, as shown above, the presumably more hydrophilic hMNP exhibited a higher blood clearance rate as compared to the similarly sized but less hydrophilic PLGA NP. At the same time, as shown by IVM, both NP have a considerable and simultaneous access to the 4T1 tumor and their accumulation areas appeared to almost overlap. Interestingly, considerable co-accumulation in the same tumor model was observed by Miller et al. [17] for the less similar diagnostic IONP coated with carbohydrate (ferumoxytol) and therapeutic PEG-PLGA NP (ferumoxytol NP: size 19 nm, zeta potential −17 mV, T_1/2_ 70 min; PEG-PLGA NP: size 90 nm, zeta potential −33 mV, T_1/2_ 55 min). Although direct comparison of the results is not possible due to the different experimental designs of these studies, it appears that at least in this tumor model the dynamic EPR effect is not so selective in relation to the above NP parameters. Together with the previous findings of Miller et al. [17] who observed colocalization of the NP on the macroscopic but not single-cell level, the obtained results strongly suggest that the most important parameter for a successful theranostic pair is the ability of both nanoparticle agents to accumulate in the same ROIs of the peritumoral area or highly vascularized tumor regions rather than the direct overlap of the nanoparticle fluorescent signals within the single cell.

Defining the proper administration regimen and the period of maximum NP accumulation at the tumor site is essential for future translation of the formulation to clinical practice. In particular, according to the MRI data, the accumulation peak for the hMNP was found to be around 5 h. The subsequent IVM performed 5 and 1 h after the administration of the hMNP and PLGA NPs, respectively, demonstrated a similar accumulation pattern of the NP. Thus, it may be expected that tumors with high hMNP accumulation in MRI scans would be responsive to the subsequent treatment with therapeutic PLGA NP. It is worth mentioning that in contrast to MRI, IVM allows to directly observe the NP extravasation behavior and distinguish between the local perivascular accumulation of NP and their deeper penetration into the tissues, which can dramatically influence the potential efficacy of nanotherapeutics [18,70]. In this regard, understanding the NP distribution pattern on the cellular level from the moment they enter the bloodstream and improving the methods of colocalization analysis can help to optimize the dual theranostic design as well as to develop appropriate administration protocols.

The strategy of dual theranostic agents can probably be used systematically to monitor the patients’ predisposition to accumulate nanotherapeutics during chemotherapy. However, more experiments with repeated administration of diagnostic and therapeutic NP are required to provide additional evidence on the applicability of this approach. It is important to further investigate whether the hMNP will demonstrate the distribution behavior similar to other nanoformulations. It is generally known that the NP microdistribution and tissue penetration depend significantly on their size, surface properties, and shape [71]. At the same time, as reported previously, ferumoxytol could predict the ability of polymeric NP to accumulate in the tumor even despite the significant differences in the circulation half-lives and microdistribution (predominant ferumoxytol accumulation in TAM) of the formulations [17]. Understanding how similar the properties of the diagnostic and therapeutic NPs need to be for their effective performance as a theranostic pair is key to addressing this issue.

In this context, the real-time analysis that could reveal the correlations between the colocalization patterns of the two NP types appears to be helpful for the selection of an appropriate “diagnostic companion” for different nanoparticle-based anti-cancer agents. The analysis of the leakage area may possibly be applied in early preclinical studies to identify the prognostic potential of the diagnostic NP (based on the extent of their colocalization with the therapeutic NP) and determine the overall efficacy of the theranostic system (based on the tumor leakage characteristics and the ability of both NP to penetrate into the tumor tissue and peritumoral area).

## 4. Materials and Methods

### 4.1. Materials

Resomer^®^ RG 502 H (poly(D,L-lactide-co-glycolide), acid terminated, MW 7000–17,000 Da) was a gift from Evonik Corporation (Darmstadt, Germany). Doxorubicin hydrochloride was obtained from Teva Pharmaceutical (Tel Aviv-Yafo, Israel). Cyanine3 (Cy3) amine and Cyanine5 (Cy5) amine were from Lumiprobe (Moscow, Russia). Cross-linker divinyl sulfone (DVS) was from Alfa Aesar (Yehud, Israel). Fluorescein isothiocyanate (FITC), Cyanine3 (Cy3) NHS ester and human serum albumin (HSA) were purchased from Sigma (Darmstadt, Germany). All other reagents were of the highest grade available. The 4T1 murine mammary carcinoma cell line was purchased from the American Type Culture Collection (ATCC, Manassas, VA, USA).

### 4.2. Preparation of Fluorescently Labeled Hybrid Ce^3/4+^-Doped Maghemite NPs Encapsulated in HSA Matrix (hMNP)

The hMNP were prepared by a two-step procedure as described previously in [28]. Firstly, the doped iron oxide MNPs were obtained by Massart’s basic co-precipitation of two types of Fe^2/3+^ salts followed by oxidation using cerium ammonium nitrate (CAN) and surface modification (doping) with a (CeL_n_)^3/4+^cation/complex [72]. The doped CAN-MNP formed a stable suspension with an average particle size of 6.61 ± 2.04 nm (by TEM; Tecnai G2 microscope, FEI –Teramo fisher, Hillsboro, OR, USA) and a positive charge of +45.7 mV. The second step involved encapsulation of the CAN-MNP in a fluorescently labeled HSA matrix. To prepare the fluorescently labeled HSA, the reactive derivatives of the fluorescent dyes—fluorescein isothiocyanate (FITC) or Cyanine3 (Cy3) NHS ester—were added to a solution of 50 mg of HSA in 0.55 mL of a sodium bicarbonate solution (pH = 8.5, dye/HSA mass ratio = 1:500) followed by constant stirring overnight at room temperature or at 4 °C for Cy3 and FITC, respectively. Next, the suspension of the doped CAN-MNP was added to the solution of fluorescently labeled HSA (Fe/HSA mass ratio = 1:50) and the mixture was stirred for 30 min. Then, 3.86 mL of ethanol (anti-solvent) was added at once to the reaction, supplementing a final concentration of 10 mg/mL with additional 30-min stirring. Thereafter, a divinyl sulfone (DVS) solution (140 µL, 5% *w*/*w* in EtOH) was added, and the mixture was stirred for 1 h at 55 °C. The resulting stable suspensions of the fluorescently labeled Cy3-hMNP and FITC-hMNP were purified by washing with ddH_2_O (3 × 10 mL) using the Vivaspin centrifugal filter device (100 K, 4000 rpm, 20 min, 18 °C; Merck KGaA, Darmstadt, Germany) and re-dispersed in ddH_2_O (20 mL).

### 4.3. Preparation of Doxorubicin-Loaded PLGA Nanoparticles Labeled with Cyanine5 (Dox-PLGA-Cy5)

Dox-PLGA-Cy5 nanoparticles were obtained by the double emulsion—solvent evaporation technique (*w/o/w*) using a polymer that was preliminarily coupled with Cy5 amine derivative (synthesis described in Appendix A). Doxorubicin hydrochloride was dissolved in 0.001 N HCl, and this solution was added to a PLGA solution in dichloromethane followed by emulsification using a high-shear homogenizer (Ultra-Turrax T18 Basic, IKA-Werke, GmbH, Staufen, Germany). The obtained primary emulsion (*w*/*o*) was added to a 1% solution of polyvinyl alcohol in phosphate buffered saline (PBS, 0.01 M, pH = 7.4) and emulsified using first a high-shear homogenizer (Ultra-Turrax T18) and then by high-pressure homogenization (Microfluidizer M-110P, Microfluidics, Newton, MA, USA). Dichloromethane was evaporated in vacuo; the resulting suspension was filtered through a sintered glass filter and freeze-dried with addition of 5% (*w*/*v*) of cryoprotectant D-mannitol. The freeze-dried nanoparticles were stored at 4 °C.

Additionally, the fluorescent placebo nanoparticles without doxorubicin (PLGA-Cy5 nanoparticles) were produced using a single *o/w* emulsion technique: PLGA was dissolved in dichloromethane and added to a PVA aqueous solution; the mixture was emulsified first using a high-shear homogenizer and then using high-pressure homogenization. All the following steps were as described for doxorubicin-loaded nanoparticles.

### 4.4. Nanoparticle Characterization

#### 4.4.1. Size, Polydispersity Index (PDI) and Zeta-Potential

The average size and polydispersity index (PDI) of all nanoparticles were measured by dynamic light scattering (DLS) using the Zetasizer NanoZS (Malvern Instruments, Malvern, UK; 5 mW He-Ne laser, operating wavelength 633 nm, 20 °C). The zeta-potential was determined using the same instrument by electrophoretic light scattering (ELS) in a dip DTS1060C-Cleare ζ cell. All measurements of the hMNPs were performed in triplicates with 50-fold dilution in ddH_2_O.

#### 4.4.2. Dox-PLGA-Cy5 Nanoparticles: Evaluation of Drug Loading, Encapsulation Efficiency, and Drug Release Rate

These properties were analyzed spectrophotometrically as described previously [44]. In brief: to determine the drug loading, samples were dissolved in DMSO and the concentration was measured using a calibration curve. For encapsulation efficiency, nanoparticles were resuspended in distilled water and centrifuged. Supernatants were assayed for unbound doxorubicin concentration using a calibration curve. The drug encapsulation efficiency was calculated as the ratio between the bound drug (determined as the difference between the total and unbound doxorubicin content) and the total drug concentration

#### 4.4.3. Drug Release Studies

The in vitro release of doxorubicin from the Dox-PLGA-Cy5.5 nanoparticles was evaluated in PBS at pH 7.4 and 4.5. The freeze-dried nanoparticles were resuspended in PBS to sink conditions, or 4.5, and were incubated at 37 °C in an orbital shaker. At certain predetermined time points (0, 1, 2, 3, 4, 6, and 24 h), 1.5 mL aliquots were obtained and centrifuged, and the nanoparticles were separated by centrifugation (100,000× *g* for 30 min, Avanti JXN-30 Centrifuge System (Beckman Coulter, Brea, CA, USA). The concentration of doxorubicin in the supernatant was measured spectrophotometrically at 480 nm.

#### 4.4.4. Quantitative Analysis of Elemental Composition of the hMNP

The Fe concentration analysis was performed by inductively coupled plasma optical emission spectrometry (ICP-OES) using the Spectro Arcos FHX22 MultiView plasma spectrometer ( SPECTRO Analytical Instruments GmbH, Kleve, Germany). The samples were dissolved in concentrated HCl (≈ 0.35 mL) at ambient temperature and ddH2O dilution.

The ultraviolet–visible (UV–Vis) spectra were obtained on the Cary 100 Bio UV–Vis spectrometer (Agilent Technologies, Santa Clara, CA, USA). The samples were dispersed in water (ca. 0.1 mg/mL).

#### 4.4.5. Fluorescence Spectroscopy

The fluorescence spectra were obtained on the Cary Eclipse Fluorescence spectrometer (Agilent Technologies, Santa Clara, CA, USA). The samples were dispersed in water (ca. 0.05 mg/mL). Optical properties (quantum yield, absorption coefficient, brightness) were evaluated as described in [45]. The fluorescence quantum yields of the obtained NPs, both therapeutic and diagnostic, were measured following the recommendations of IUPAC [73] by the relative method [74] as previously described in detail in [45]. Absorption coefficient and brightness were also evaluated.

#### 4.4.6. Scanning Electron Microscopy (SEM) of PLGA NP

The freeze-dried PLGA NP were resuspended in distilled water and centrifuged at 20,000 rpm (14 °C, 30 min) to remove D-mannitol and excess PVA. Then the supernatant solution was discarded, and the precipitated nanoparticles were resuspended in the same amount of distilled water using Vortex (Velp Scientifica, Deer Park, NY, USA) and ultrasound bath (Bandelin Electronic GmbH & Co. KG, Berlin, Germany). The drop of PLGA NP suspension (diluted 5-fold) was placed on a sample table, air-dried, and then sputtered with a layer of platinum for 6 to 30 s which produces a surface layer of Pt with a thickness of 13 nm. The table was then placed in the JSM-7600F Schottky field emission scanning electron microscope (JEOL, Nieuw-Vennep, Japan), and the sample was imaged in the secondary electron mode (planar). Capturing mode: high vacuum, accelerating voltage up to 15 kV, detection of secondary electrons (planar).

#### 4.4.7. Transmission Electron Microscopy (TEM) of hMNP and PLGA NP

For the hMNP investigation by TEM, a FCF 400 grid was glow discharged with the EmiTech K100 machine; then 5 µL of the sample was loaded on the grid. After 1 min, the sample was blotted and the access material was removed; then, 5 µL of uranyl acetate was loaded for 30 s, and then blotted, washed with DD water, and air dried. The sample was then inspected with the Tecnai G2 microscope (FEI—Teramo fisher) with an acceleration voltage of 120 kV. The images were obtained using a Digital Micrograph with the Multiscan Camera model 794 (Gatan) in different resolutions. The PLGA NP samples were stained with uranyl acetate. The images were obtained using the JEOL JEM-1400 microscope (JEOL, Nieuw-Vennep, Japan) at 120 kV accelerated voltage.

### 4.5. In Vitro Experiments

#### 4.5.1. 4T1 Murine Mammary Carcinoma Cells

The 4T1 murine mammary carcinoma cells were purchased from the American Type Culture Collection (ATCC, Manassas, VA, USA). The cells were cultured in RPMI medium (Gibco) supplemented with antibiotics (100 U/mL penicillin, 100 g/mL streptomycin, Gibco), L-glutamine (2 mM, Gibco), and 10% fetal bovine serum (Biowest, Lakewood Ranch, FL, USA). The cells were grown under standard culture conditions (37 °C and 5% CO_2_). The 4T1mScarlet cells were derived from the initial 4T1 cell line via lentiviral transduction [75]. The mScarlet-positive cells were sorted with the MoFlo cell sorter (Beckman Coulter, Brea, CA, USA).

#### 4.5.2. Investigation of Nanoparticle Internalization by 4T1 Tumor Cells In Vitro

The nanoparticle internalization pathways and intracellular distribution were investigated by confocal laser scanning microscopy (CLSM). The 4T1 (4T1 or 4T1-mScarlet) cells (10^6^ cells/well) were added to the polymeric coverslip-bottom 35 mm confocal dishes (Ibidi) and allowed to attach overnight. After 24 h, the hMNP labeled with Cy3 or FITC were added in cell culture medium (RPMI) at a final concentration of 200 µg/mL (MNP) and incubated for 40 min. The 4T1 cells were also incubated with the hMNP (final concentration 200 µg/mL) and PLGA-Cy5 NP (100 µg/mL) simultaneously. Then the cells were washed with PBS and the lysosomes were stained with Lysotracker Green DND26 according to the manufacturer’s protocol. In several experiments, the cell lysosomes and late endosomes were stained with CellLight™ Lysosomes GFP and CellLight™ Late endosomes-RFP (BacMam 2.0), respectively, according to the manufacturer’s protocol (Invitrogen, Waltham, MA, USA).

To investigate the main mechanism of nanoparticle internalization, the cells were preincubated with the inhibitors (1 h preincubation) of the main endocytic pathways: methyl-ß-cyclodextrin (MβCD, 2.5–5mM) and nystatin (20–70 μM) for caveolin-mediated endocytosis, and chlorpromazine (40 μM) for clathrin-mediated endocytosis, and macropinocytosis inhibitor for amiloride (7.5–60 μM). Dynasore (80 μM) was used to inhibit both caveolin- and clathrin-mediated endocytosis pathways. Then, the cells were washed with PBS and the nanoparticle uptake was measured as described above. Two NP concentrations were tested—50 μg/mL and 100 μg/mL. The concentration ranges of endocytosis inhibitors were chosen based on the primary cytotoxicity experiments and literature data [56,76]. Endocytosis inhibitor concentrations up to 40, 120, 160, 100 μM, and 5 mM for chlorpromazine, amiloride, dynasore, nystatin, and MβCD, respectively, were found to be nontoxic. The images were acquired using the Nikon confocal microscopy system Nikon A1R MP+ (Nikon Instruments, Tokyo, Japan); maximum projections along the Z-axis were made using NIS- Elements AR software. Lasers with emission wavelengths of 405 nm, 488 nm, 561 nm, and 638 nm were used.

#### 4.5.3. Evaluation of hMNP Hemocompatibility and Cytotoxicity

The hemolytic activity of hMNP was evaluated ex vivo using a colorimetric method according to the technique described in [57] with modifications. The human blood samples were kindly provided by the N.N. Blokhin Cancer Research Center of the Russian Ministry of Health (Moscow, Russia).

In brief, the blood samples were centrifuged (Eppendorf 5804R centrifuge; 900× *g*, 10 min, +18 °C) and the RBCs were then resuspended in PBS to a final concentration of 4∙10^9^ cells/mL. A series of hMNP dilutions (0.195–100 μg/mL) were prepared and added to the RBCs samples followed by a 2 h incubation at +37 °C in a shaker incubator. After incubation, samples were centrifuged (900× *g*, 5 min, +18 °C), and 100 μL aliquots of the supernatant were transferred to 96-well plates. Sodium dodecyl sulfate (final concentration 0.06%) was added to the experimental samples to induce hemichrome formation. The amount of hemoglobin in the sample was determined by evaluating the optical density of the solution at λ_ex_ = 540 nm using the EnSpire plate analyzer (PerkinElmer, Waltham, MA, USA). The percentage of hemolysis was calculated as the ratio of the optical density of the sample to the optical density of the positive control (×40), n = 5.

The nanoparticle cytotoxicity was assessed by the commonly used MTS assay (Protocol TB245, Promega). The CellTiter 96 AQueous One Solution reagent (Promega) was used. Briefly, the hMNP samples were incubated with 4T1 cells in the concentration range of 6.25–400 μg/mL (as Fe) for 24 h. The absorbance was measured using the plate reader PerkinElmer EnSpire (Waltham, MA, USA). The cell viability was calculated as a percentage compared to the control (untreated cells).

### 4.6. In Vivo Experiments

#### 4.6.1. Tumor Models

Animal experiments were approved by the Ethics committee of V. Serbsky Federal Medical Research Center of Psychiatry and Narcology (Moscow, Russia). Female BALB/c mice were obtained from Andreevka Animal Production Center (Andreevka, Russia). The orthotopic 4T1 tumor model was established by injection of 4T1 cells or 4T1mScarlet cells expressing a mScarlet fluorescent protein (1 × 10^6^ cells) into the mammary fat pad of Balb/c mice. The heterotopic tumors were obtained by subcutaneous injection of 4T1 (or 4T1-mScarlet) cells (1 × 10^6^) into the hind flank.

#### 4.6.2. Determination of T2 Relaxivities for Magnetic Nanoparticles

The T2 relaxivity was determined by MRI using the ClinScan 7T MR-tomograph (Bruker Biospin, Billerica, MA, USA). The MR images of tubes containing sample solutions with the known concentrations of Fe^3+^ were obtained by a Spin Echo (SE) sequence with the following parameters: TR = 10,000 ms, TE = 8, 16, 24,…, 240 ms. The signal intensity was evaluated in each test tube, and the T2 time was calculated using the following equation:$$Si=S_{0}e^{-TE/T2}$$
where S_0_—the signal at an initial time, Si—the signal at TE time.

The T2 relaxivity values were obtained from linear fitting of T2^−1^ relaxation times as a function of Fe concentration. The T2 relaxivity values were determined for the hMNP and commercially available 35-nm Absolute Mag™ Carboxyl Dextran Magnetic Particles labeled with Rhodamine B (CD Bioparticles, USA).

#### 4.6.3. Magnetic Resonance Imaging

Investigation of the hMNP distribution and accumulation in the 4T1 tumor-bearing BALB/C mice by MRI was performed using the Clinscan 7T MR tomograph (Bruker) equipped with a 20-cm volumetric coil as a transmitter and a 4-segment surface coil as a radio frequency signal detector. The tumor-bearing BALB/C mice were anesthetized with 2% isoflurane and scanned before and 1 h, 5 h, and 24 h after the i.v. injection of hMNP (~3.5 mg/kg as Fe) on day 10 after tumor inoculation. The T2-weighted turbo spin echo regimen (TR = 2000 ms, TE = 42 ms, slice thickness = 1 mm, base resolution 380 × 640) was used for image acquisition of coronal planes. The T2*-weighted gradient echo regimen (TR = 400 ms, TE = 3.51 ms, slice thickness = 0.7 mm, FOV = 31.25 × 40 mm, base resolution 256 × 200) was used for image acquisition of transverse planes.

#### 4.6.4. Intravital Microscopy

Evaluation of the in vivo nanoparticle distribution by intravital microscopy (IVM) was performed in Balb/c mice with the subcutaneous 4T1mScarlet tumor on day 8–10 after tumor inoculation. The mice were anesthetized with ketamine and xylazine. To obtain access to the subcutaneous tumor, a small midline incision along the spine from 5 mm above the tail base to near the shoulder apex was performed and the skin was reflected. The connective tissue overlying the tumor was gently removed and edged and the skin flap was secured using sutures to expose the tumor for imaging. Imaging was performed using the inverted laser scanning confocal microscope Nikon A1R MP+ (Nikon, Japan). The PLGA NP (50 mg/kg) and hMNP (3.5 mg/kg as Fe) were administered through a catheter into the tail vein. The fluorescently-labeled CD31 antibodies were intravenously injected to stain the endothelial cells. Fluorescently-labeled CD11b and Ly6G antibodies were injected to visualize leukocyte subpopulations—neutrophils (CD11b+ Ly6G+) and monocytes (CD11b+ Ly6G-).

The IVM study was performed within the first minutes (0–40 min), 2, 6, and 24 h after the NP administration. The NP localization was detected by imaging a series of sections along the z-axis. Different tumor regions (5 to 10 selected spots) were scanned sequentially using 405 nm, 488 nm, 561 nm, and 647 nm lasers for 60–120 min (1 frame [512 × 512 or 1024-1024 pix]/40–60 s). Then, the 3D-reconstruction of the z-stacks was performed. Plan Apo 20×/0.75 DIC N and Apo LWD 40×/1.15 S objectives (Nikon, Japan) were used for IVM. After the imaging, the animals were euthanized and the liver and spleen were harvested for additional confocal imaging.

The blood circulation half-lives of the NP were determined by analyzing the fluorescence intensity profiles in multiple vessels (several fields of view [FOV]) within the first 60–80 min after the NP administration, as described in [63].

To analyze the colocalization between two nanoparticle types within the time frame, the nanoparticle microdistribution and accumulation were investigated using different administration protocols with the intervals of 3 h, 5 h, and 24 h between the doses: (1) IVM was performed immediately after simultaneous administration of the PLGA-Cy5 NP and hMNP-FITC (imaging was performed for up to 120 min); (2) IVM was performed 2 h after the hMNP-FITC injection and 1 h after the PLGA-Cy5 NP injection (3 h/1 h); (3) IVM was performed 4 h after the hMNP-FITC injection and 1 h after the PLGA-Cy5 NP injection (5 h/1 h); (4) IVM was performed 23 h after the hMNP-FITC injection and 1 h after the PLGA-Cy5 NP injection (24 h/1 h). Each group contained 8 animals.

Each frame obtained during IVM was analyzed using the NIS Elements AR software. To analyze the NP microdistribution pattern and determine leakage areas for both NP types upon different administration regimens, the binary areas were generated for each frame using a fluorescence intensity threshold for the corresponding channels (Alexa 488 or Cy5 for the hMNP-FITC or PLGA-Cy5 NP, respectively). Random ROIs (regions of interest, 50 × 50 µm) were generated for each frame (3–9 ROIs per frame). The leakage areas were determined for each frame (10–15 frames) for 5–10 selected spots. To compare the spatial distribution of the PLGA NP and hMNP (leakage areas and localization), the following parameters were introduced: S (Binary area, µm^2^)/ROI (2500 µm^2^), %. The administration regimen with both NP types injected simultaneously was used as a positive control for high colocalization between the NP. The regimen with the PLGA NP administered 24 h after the hMNP injection was used as a negative control.

The frequency of NP distribution to the same ROI was analyzed based on the fluorescence signal of NPs within the randomly selected ROIs upon different administration regimens (at least 100 ROIs were analyzed for each administration regimen).

Doxorubicin fluorescent signal is generally weak in IVM studies. To address this issue, the signal processing was performed (IA module, NIS Elements—AI denoise) to improve visualization of doxorubicin distribution as well as to remove the background noise from tissue. Afterwards the doxorubicin fluorescent channel was contrast-enhanced to determine its precise localization. In particular, the described processing was applied to Figure 11.

### 4.7. Statistical Analysis

The Student’s *t*-test and ANOVA (one or two-way) followed by Bonferroni’s, Tukey’s, or Sidak’s multiple comparison test were used for data analysis. The differences were considered significant at values of *p* < 0.05. GraphPad Prism 9 software was used.

## 5. Conclusions

Complex evaluation of the model theranostic pair—hMNP and Dox-PLGA NP—described in the present paper involved all steps from the NP synthesis to the detailed analysis of their intratumoral and peritumoral microdistribution and colocalization. Due to the high spatial resolution of IVM, the NP microdistribution in the vasculature of tumor and peritumoral space could be followed starting from the first minutes post injection and further on—through all steps of the NP extravasation from their attachment to the vessel walls to micro- and macroleakage formation. The NP extravasation via neutrophil-associated leakages confirmed the role of immune cells (neutrophils) in the NP penetration into tumor tissue. Furthermore, IVM demonstrated the efficient delivery of the NP therapeutic payload (doxorubicin) to tumor tissue. Thorough analysis of the leakage areas indicated the high incidence of the NP co-distribution to the same regions for various administration regimens, suggesting that the hMNP and PLGA NP could be a promising theranostic pair. The suggested analysis of the microdistribution pattern and colocalization for two NP types can be used in the preclinical studies for the selection of appropriate “companion diagnostics” for different nanotherapeutics.

Known heterogeneity of tumors suggests that further studies should focus on the investigation of various, clinically relevant tumor models to better understand the requirements for nanoparticle parameters and potential advantages and limitations of this approach.

## Figures and Tables

**Figure 1 ijms-24-00627-f001:**
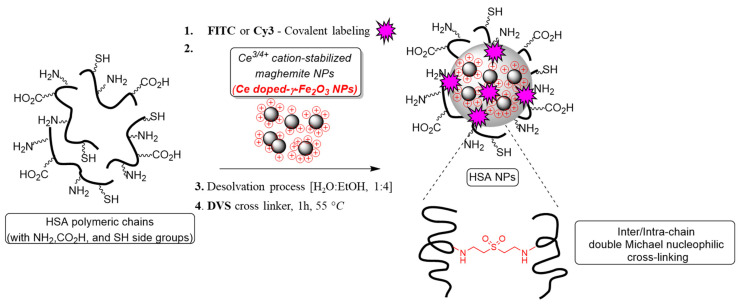
Schematic representation of the hMNP preparation procedure.

**Figure 2 ijms-24-00627-f002:**
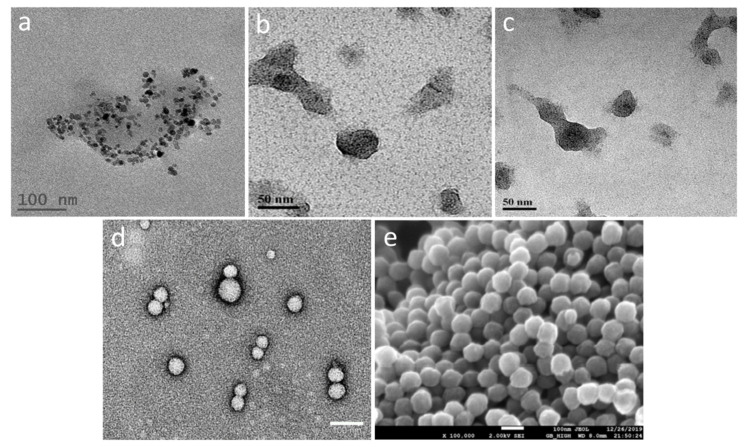
Micrographs of PLGA-Cy5 NP and hMNP-FITC obtained by SEM and TEM: (**a**) TEM image of hMNP without staining; TEM images of hMNP-FITC (**b**) and hMNP-Cy3 (**c**) stained with uranyl acetate (Tecnai G2 microscope, FEI –Teramo fisher); (**d**) TEM images of PLGA-Cy5 NP (JEOL JEM-1400); (**e**) SEM image of PLGA-Cy5 NP (sputter-coated with Pt, JSM-7600F Schottky field emission scanning electron microscope, JEOL, Japan).

**Figure 3 ijms-24-00627-f003:**
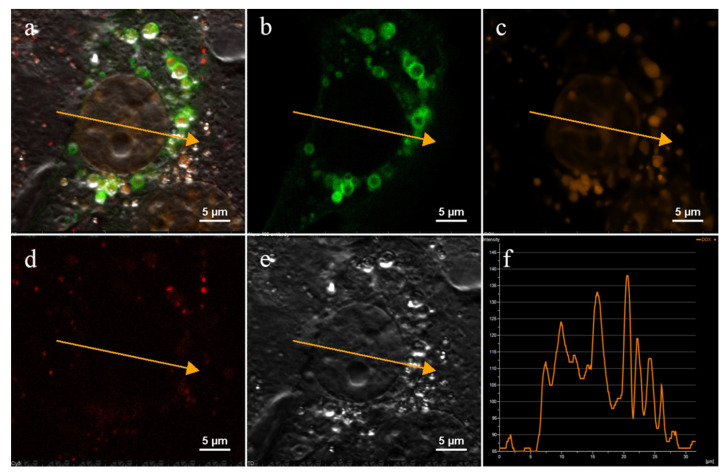
Intracellular localization of doxorubicin-loaded PLGA-Cy5 NP (Dox-PLGA-Cy5 NP) in 4T1 cells (60 min of incubation). (**a**) Merged image; (**b**) lysosomes—CellLight™ Lysosomes GFP, BacMam 2.0 (green channel); (**c**) Doxorubicin (orange channel); (**d**) PLGA-Cy5 NP (red channel); (**e**) cell contours—differential interference contrast (DIC); (**f**) fluorescence intensity profile along the vector (orange arrow), X-axis—µm, Y-axis—fluorescence intensity (standard units). CSLM. Scale bar—10 μm.

**Figure 4 ijms-24-00627-f004:**
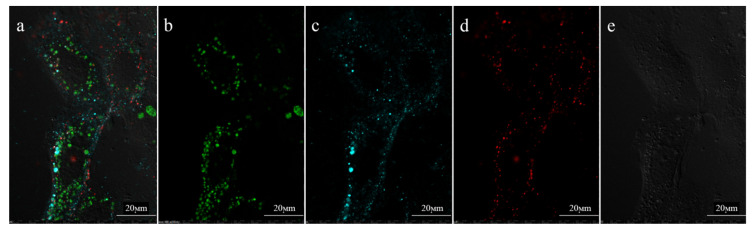
Internalization of hMNP-Cy3 and PLGA-Cy5 NP into 4T1 cells (40 min of incubation): (**a**) merged image; (**b**) lysosomes (Lysotracker Green DND26); (**c**) hMNP-Cy3; (**d**) PLGA-Cy5 NP; (**e**) cell contours—differential interference contrast (DIC). Scale bar—10 μm.

**Figure 5 ijms-24-00627-f005:**
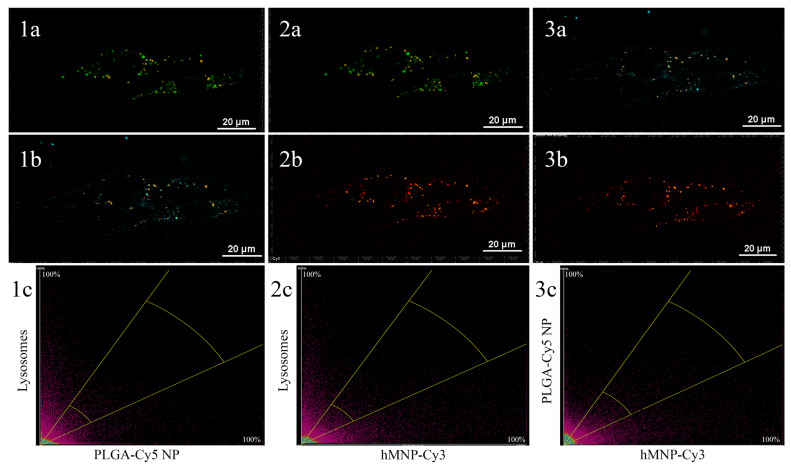
Investigation of nanoparticle colocalization in 4T1 cells (deconvolved image, 40 min of incubation). (1)—Lysosomes (**1a**)—PLGA-Cy5 NP (**1b**), (2)—Lysosomes (**2a**)—hMNP-Cy3 (**2b**), (3)—PLGA-Cy5 NP (**3a**) and hMNP-Cy3 (**3b**), (**1c**–**3c**) colocalization scatterplots. Applied synthetic colocalization channel is depicted in orange. Scale bar—20 μm.

**Figure 6 ijms-24-00627-f006:**
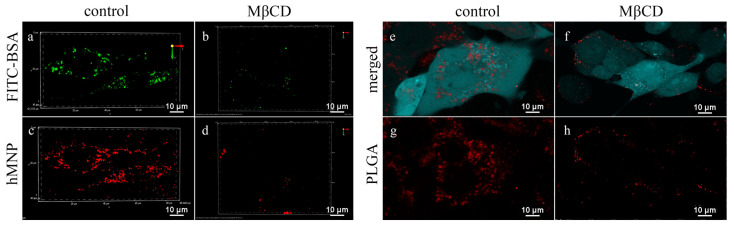
Investigation of hMNP-Cy3 and PLGA-Cy5 NP uptake mechanism by 4T1-mScarlet cells (60 min of incubation). (**a**) Control—FITC-BSA. (**b**) FITC-BSA internalization into cells treated with MβCD. (**c**) hMNP-Cy3 internalization into untreated cells (control). (**d**) hMNP-Cy3 internalization into cells treated with MβCD. Scale bar—20 μm. (**e**) Merged image 4T1-mScarlet cells + PLGA-Cy5 NP. (**f**) Merged image—PLGA-Cy5 NP + 4T1-mScarlet cells treated with MβCD. (**g**) PLGA-Cy5 NP internalization into untreated cells. (**h**) PLGA-Cy5 NP internalization into cells treated with MβCD. Scale bar—10 μm.

**Figure 7 ijms-24-00627-f007:**
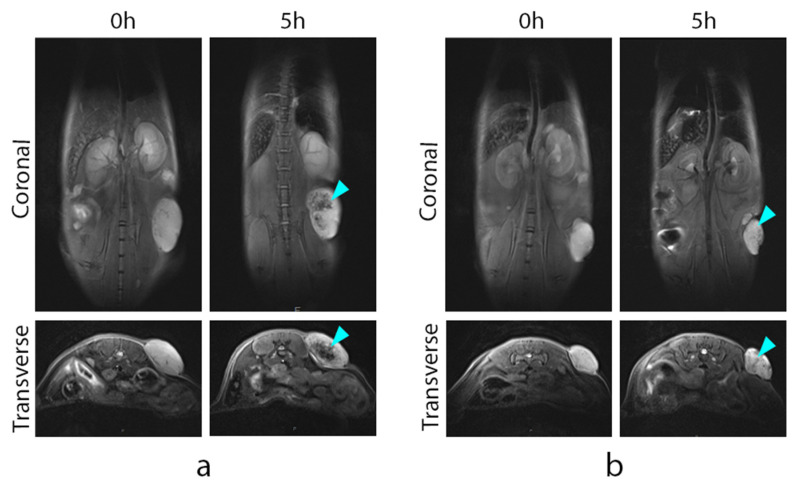
Representative T2-weighted images of mice with subcutaneously implanted 4T1 tumor before (0 h) and 5 h after the intravenous injection of the hMNP (**a**) and Absolute Mag™ nanoparticles (CD-MNP) (**b**) (3.5 mg/kg Fe). Negatively contrasted tissues indicate the hMNP and CD-MNP accumulation (blue arrowheads). Coronal and transverse projections are demonstrated.

**Figure 8 ijms-24-00627-f008:**
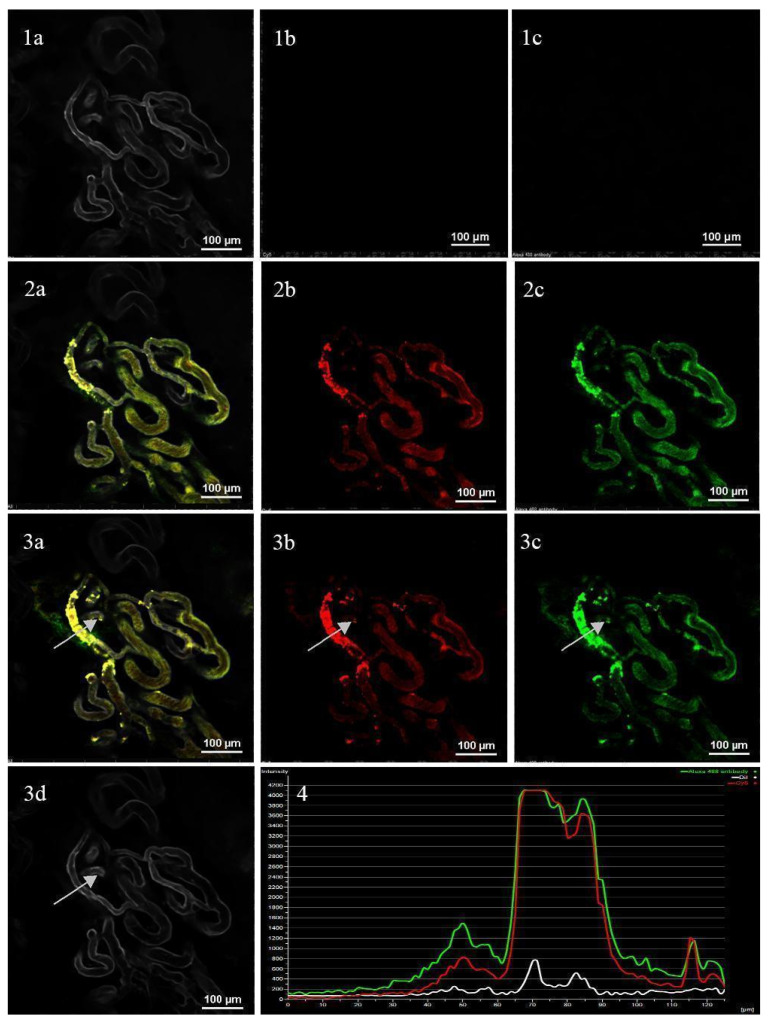
IVM study of hMNP-FITC and PLGA-Cy5 NP distribution in tumor vasculature and peritumoral area in 4T1 tumor—bearing mice upon simultaneous administration (0–45 min after NP injection). The white arrow points to the extravascularly located NP (microleakage). Line 1 (**1a**–**1c**)—1 min of PLGA-Cy5 NP exposition; line 2 (**2a**–**2c**)—15 min; line 3 (**3a**–**3d**)—30 min. (**1a**–**3a**) Merged image (hMNP-FITC and PLGA-Cy5 + CD31-PE); (**1b**–**3b**) PLGA-Cy5 NP (red channel); (**1c**–**3c**) hMNP-FITC (green channel); (**3d**) CD31-PE; **4**—fluorescence intensity profile along the vector (white arrows in images 3a–d). X-axis—µm; Y-axis—fluorescence intensity (standard units); the colors correspond to the channels. Scale bar—20 μm.

**Figure 9 ijms-24-00627-f009:**
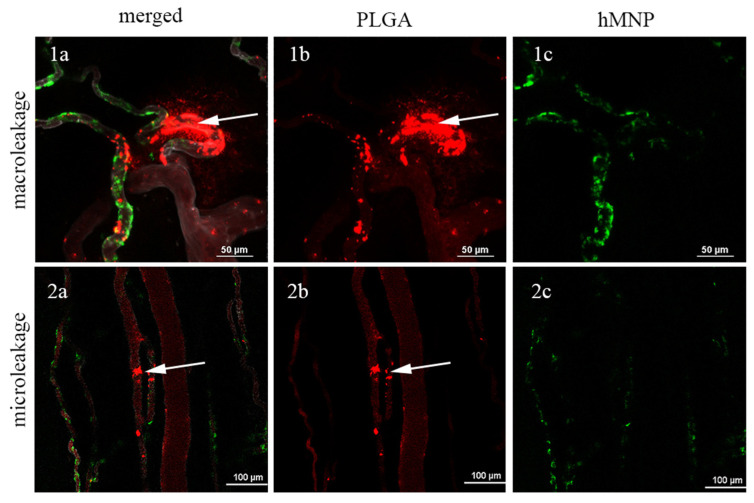
Investigation of nanoparticle extravasation in tumor and peritumoral area in 4T1 tumor-bearing mice. Representative images of macroleakages (**1**) and microleakages (**2**): (**1a**,**2a**) merged images; (**1b**,**2b**) PLGA-Cy5 NP (red channel); (**1c**,**2c**) hMNP-FITC (green channel). Leakage areas are indicated by white arrows. Scale bar—50 μm (**1**) and 100 μm (**2**).

**Figure 10 ijms-24-00627-f010:**
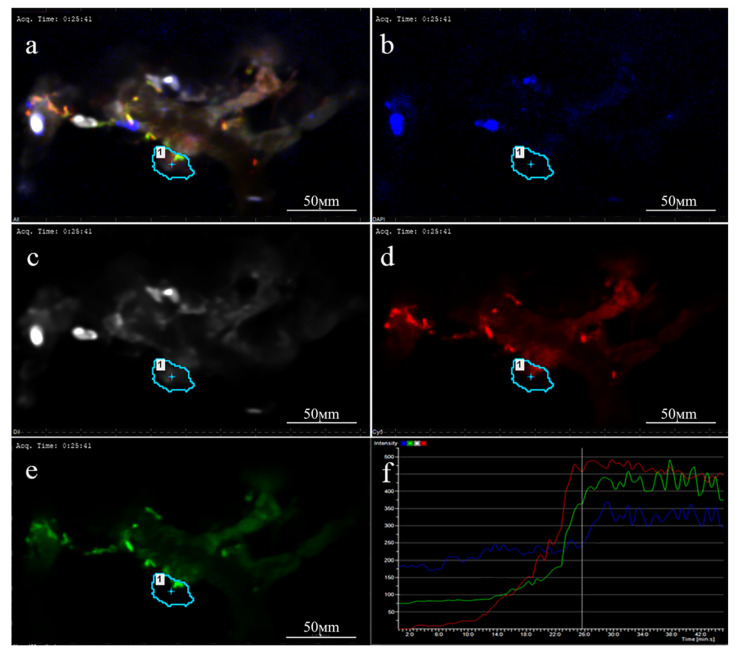
Extravasation of hMNP and PLGA NP through macroleakages in tumor vasculature and peritumoral space in 4T1 subcutaneous tumor (0–45 min after injection of both NP). (**a**) Merged image; (**b**) Ly6G-positive neutrophils; (**c**) CD45-PE; (**d**) PLGA-Cy5 NP; (**e**) hMNP-FITC; (**f**) fluorescence intensity profiles (Y-axis) measured in dynamics. Scale bar—50 μm.

**Figure 11 ijms-24-00627-f011:**
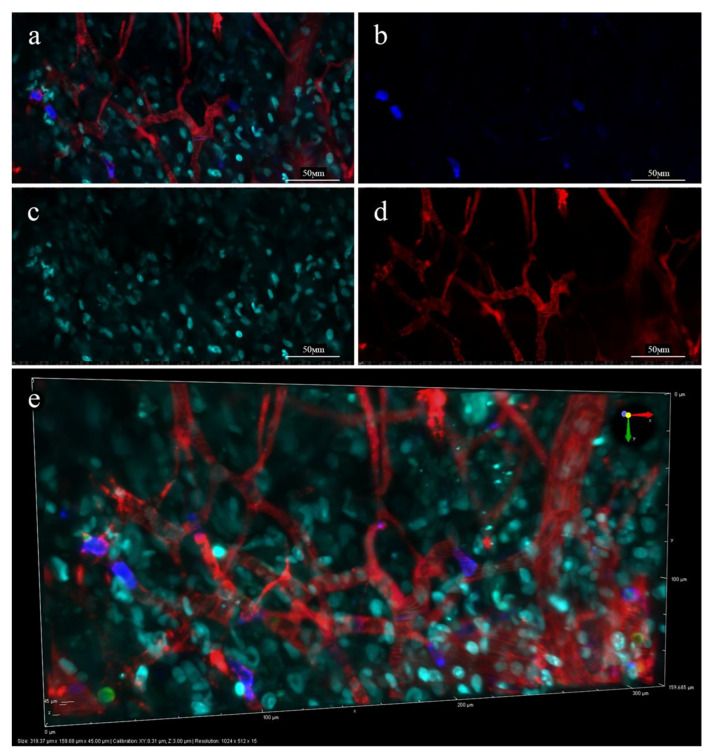
Investigation of Dox-PLGA-Cy5 microdistribution in 4T1 subcutaneous tumor. Representative IVM images. (**a**) Merged image; (**b**) Ly6G-positive neutrophils; (**c**) Dox; (**d**) PLGA-Cy5 NP; (**e**) 3D reconstruction of peritumoral tissue fragment. Scale bar—50 μm.

**Figure 12 ijms-24-00627-f012:**
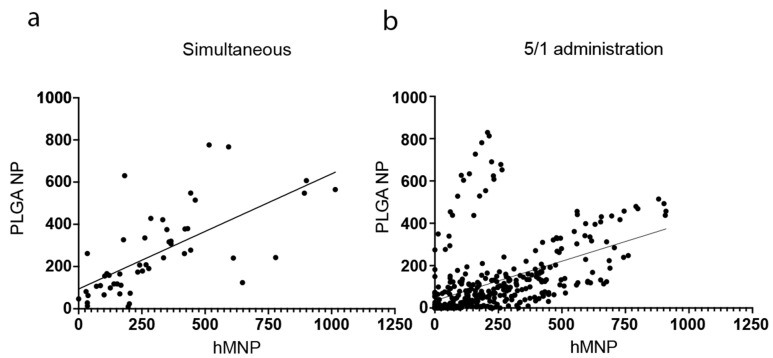
Analysis of the leakage areas for both NP types after different administration regimens. Each point on the graphs represents ROI containing information on the areas (μm^2^) occupied by the NP. (**a**) Analysis of NP distribution upon simultaneous administration. (**b**) Analysis of NP distribution with 5 h/1 h administration regimen. *p* = 0.0495.

**Table 1 ijms-24-00627-t001:** Physicochemical parameters of fluorescently labeled PLGA NP and hMNP.

Nanoparticle Type	Hydrodynamic Diameter * (nm)	PDI	Zeta Potential (mV)	Drug Loading (%)	Drug Encapsulation Efficiency (%)
PLGA-Cy5	96 ± 1	0.146 ± 0.027	−30.1 ± 0.3	-	-
PLGA-Cy3	96 ± 1	0.117 ± 0.011	−33.4 ± 1.5	-	-
Dox-PLGA-Cy5	88 ± 1	0.073 ± 0.014	−14.4 ± 0.1	7.6 ± 0.1	96.6
hMNP-FITC	133 ± 1	0.108 ± 0.008	−36.3 ± 0.5	-	-
hMNP-Cy3	144 ± 1	0.108 ± 0.014	−37.6 ± 0.5	-	-
hMNP	109.3 ± 2.5	0.133 ± 0.029	−38.2 ± 2.2	-	-

* evaluated by intensity.

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
