# Peer review of "Supermagnetic Human Serum Albumin (HSA) Nanoparticles and PLGA-Based Doxorubicin Nanoformulation: A Duet for Selective Nanotherapy"

_ijms, 2022, doi:10.3390/ijms24010627_

Round 1

Reviewer 1 Report

The author presented the paper "Supermagnetic human serum albumin (HSA) nanoparticles and PLGA-based doxorubicin nanoformulation: a duet for selective nanotherapy"

1) Many more 2-3 year paper have to be presented to show the perspectives of the area. Papers older than 20 years looks very strange for Introduction section. In general, the reference list is poor. It can be improved.

2) Albumin features for MNP coating are not presented in the Introduction section in an appropriate way. I recommend citing these references and to present some information.

Serum Albumin for Magnetic Nanoparticles CoatingAlbumin-Coated Single-Core Iron Oxide Nanoparticles for Enhanced Molecular Magnetic Imaging (MRI/MPI)Aggregation Properties of Albumin in Interacting with Magnetic FluidsInsights into Preformed Human Serum Albumin Corona on Iron Oxide Nanoparticles: Structure, Effect of Particle Size, Impact on MRI Efficiency, and Metabolization

3) Table 1. Can you mention that this is DLS data. Is this size, volume, or intensity mode? Please, present in SI all these data (pictures) to understand the nanoparticles' homogeneity. Usually one file with pictures is better for SI.

Have you analyzed all these conjugates by TEM?

4) Where is Table 2 with DLS data? Fig. 2 Why you present only this conjugates? Maybe the most important, the final conjugates, are better?

5) Section 2.3. Where are any initial experimental data and the results? Any cytotoxicity data for all the conjugates? It must be done before in vivo experiments. Have you studied reactive oxygen species formation of MNPs?

6) MRI. Fig. 7. It will be better to do color pictures.  Control picture without MNPs? Can you mention injection method in methods, and figure 7 caption?Can you mention the injection place in Figure 7 and 0 h point? Why you have used such injection method?

7) In your paper, you use MNP only for MRI? Have you measured magnetic properties to understand the potential for magnetic delivery? Possibly, some data about potential for hyperthermia?

8) I am a bit confused, have you done relevant cell and mouse studies about doxorubicin loaded nanoparticles?

The authors have done great work. However, the general impression is confused because of the many data about various species internationalization. The paper should be better organized with more Tables in contradistinction to text writing, synthesis schemes, etc.

Conclusion section is poor. Any explanations, the outstanding results should be presented. I don't clearly understand this highlights: "the model theranostic pair hMNP and Dox-PLGA NP in combination". This result is too general. I can write without the reading that good to combine diagnostic tool and drug. The novelty of the work should be mentioned in a better way. This will highlight the importance of the work.

Author Response

1) Many more 2-3 year paper have to be presented to show the perspectives of the area. Papers older than 20 years looks very strange for Introduction section. In general, the reference list is poor. It can be improved. 

Answer. In the Introduction we briefly touched upon the establishment and evolution of the EPR effect concept. Therefore, we considered it relevant to include the references to the original works of the pioneers of this concept – Hiroshi Maeda and Rakesh Jain (ref. 1, 9-11, 19).

The reference list was extended: altogether 22 references were added. The Introduction section now contains 25 references to the papers published in 2019-2022 (refs 2, 6, 7, 12, 14-16, 20-22, 25, 27, 29, 30, 32-39, 41-43).

 2) Albumin features for MNP coating are not presented in the Introduction section in an appropriate way. I recommend citing these references and to present some information. 

Serum Albumin for Magnetic Nanoparticles Coating, Albumin-Coated Single-Core Iron Oxide Nanoparticles for Enhanced Molecular Magnetic Imaging (MRI/MPI), Aggregation Properties of Albumin in Interacting with Magnetic Fluids,  Insights into Preformed Human Serum Albumin Corona on Iron Oxide Nanoparticles: Structure, Effect of Particle Size, Impact on MRI Efficiency, and Metabolization.

Answer. The articles suggested by the reviewer (refs 32, 33, 37, 38) as well as other references related to the role of albumin in drug delivery and for the MNP coating (refs 30, 34, 35, 36) have been added to the article. The comment is added (p. 3, paragraph 4).

3) Table 1. Can you mention that this is DLS data. Is this size, volume, or intensity mode?  

Answer: Table 1: Particle diameter was changed to Hydrodynamic diameter by intensity (nm). The comment is added in Section 2.1 (p. 4)

Please, present in SI all these data (pictures) to understand the nanoparticles' homogeneity. Usually one file with pictures is better for SI. 

Answer: The DLS information on the hMNP (both labeled and non-labeled) were added to the text. The figures presenting the results of all the DLS and Zeta-potential measurements are now added in the supporting information (Figure S1).

Have you analyzed all these conjugates by TEM?

Answer: Yes we did, the TEM images of the PLGA-Cy5 NP, hMNP, and hMNPs-Cy3 are added in Fig. 2. The non-labeled hMNP and fluorescently labeled hMNP are structurally similar. We also replaced the previous images of the hMNP-FITC composite with improved ones (Fig. 2, p. 6).

 4) Where is Table 2 with DLS data?

Answer: The reference to Table 2 in p. 6 was a mistake. There should be a reference to the data in Table 1. This error has been corrected (p.6). 

 Fig. 2 Why you present only this conjugates? Maybe the most important, the final conjugates, are better? 

The micrographs presented in Fig.2 are the final conjugates obtained after labeling; the structure of the nanoparticles was not affected and remains the same. As mentioned in the answer to comment 3, the TEM images of the PLGA-Cy5 NP, hMNP, and hMNPs-Cy3 were added in Fig. 2 (p.6).

 5) Section 2.3. Where are any initial experimental data and the results? Any cytotoxicity data for all the conjugates? It must be done before in vivo experiments. Have you studied reactive oxygen species formation of MNPs? 

Answer: The data  on the MNP cytotoxicity was added to the Results (section 2.2. p.7) and Methods (section 4.5.3). The absence of ROS generation in neutrophils in response to a slightly different type of the hybrid HSA-MNPs was evaluated in our previous study (ref. 39: Israel et al. Towards hybrid biocompatible magnetic rHuman serum albumin-based nanoparticles: use of ultra-small (CeLn)3/4+ cation-doped maghemite nanoparticles as functional shell. Nanotechnology. 2015;26(4):045601).

The corresponding graph was added to Supplementary materials (Figure S.2).

6) MRI. Fig. 7. It will be better to do color pictures.  Control picture without MNPs? Can you mention injection method in methods, and figure 7 caption? Can you mention the injection place in Figure 7 and 0 h point? Why you have used such injection method? 

Answer: The images presented in Fig. 7 (p. 14, section 2.4) are the original images obtained after MRI performance using the Clinscan 7T MR tomograph (Bruker) without any post-processing procedure. 

The nanoparticles were systemically administered by intravenous injection via the tail vein of the animals. The MRI scans obtained at 0 h point (before nanoparticles administration) is now added to Fig. 7. Systemic administration was chosen as the most relevant route that is also used for administration of the contrasting agents in clinical practice. Moreover, it enables evaluating the ability of the circulating hMNP to gradually accumulate in tumor tissues by the EPR effect. We preferred the injection in the tail vein as the less invasive technique (as compared to the retro-orbital injection).

7) In your paper, you use MNP only for MRI? Have you measured magnetic properties to understand the potential for magnetic delivery? Possibly, some data about potential for hyperthermia?

Answer. This manuscript focuses on the good imaging properties of the hMNP in MRI to be used as a diagnostic companion for the Dox-PLGA NP therapeutic nanoparticles. However, the hMNP magnetic properties are suitable for magnetic delivery and their potential for hyperthermia are important and can be used for cancer treatment.

The magnetization profiles of MNP (without albumin) were investigated in our previous studies presenting the SQUID magnetization profile with saturation magnetization  of Ms: 75.2 emu/g, as described in [ref. 26]. The magnetic delivery property of the MNPs was also investigated in-vitro and in-vivo and published on 2019. This work proves the magnetic targeting of the same MNPs attached to mTHPC by improving the selectivity and efficiency of photodynamic therapy [ref 27]. Unfortunately, no work has been done regarding hyperthermia, despite the relevance of this method of treatment with these types of particles, as can be seen in [ref 21, 22].  These references were added and discussed in the introduction section.

  1. Liron L. Israel,   Emmanuel Lellouche,   Ron S. Kenett,   Omer Green,   Shulamit Michaeli  and  Jean-Paul Lellouche. Ce3/4+ cation-functionalized maghemite nanoparticles towards siRNA-mediated gene silencing. J. Mater. Chem. B, 2014, 2, 6215–6225
  2. E Haimov-Talmoud,  Y Harel, H Schori, M Motiei, A Atkins, R Popovtzer,  J-P Lellouche and O Shefi. Magnetic Targeting of mTHPC To Improve the Selectivity and Efficiency of Photodynamic Therapy, ACS Appl. Mater. Interfaces 2019, 11, 45368−45380
  3. Agnieszka Włodarczyk , Szymon Gorgo, Adrian Rado and Karolina Bajdak-Rusinek. Review; Magnetite Nanoparticles in Magnetic Hyperthermia and Cancer Therapies: Challenges and Perspectives. Nanomaterials 2022, 12, 1807. https://doi.org/10.3390/nano12111807
  4. O. M. Lemine, Nawal Madkhali, Marzook Alshammari, Saja Algessair, Abbasher Gismelseed, Lassad El Mir, Moktar Hjiri, Ali A. Yousif  and Kheireddine El‐Boubbou. Maghemite (γ‐Fe2O3) and γ‐Fe2O3‐TiO2 Nanoparticles for Magnetic Hyperthermia Applications: Synthesis,Characterization and Heating Efficiency. Materials 2021, 14, 5691. https://doi.org/10.3390/ma14195691 

8) I am a bit confused, have you done relevant cell and mouse studies about doxorubicin loaded nanoparticles? 

The authors have done great work. However, the general impression is confused because of the many data about various species internationalization. The paper should be better organized with more Tables in contradistinction to text writing, synthesis schemes, etc.

Answer: Yes, we performed  the relevant in vitro and in vivo studies with doxorubicin-loaded Dox-PLGA-Cy5 NP. 

The investigation of Dox-PLGA-Cy5 NP internalization and intracellular trafficking and Dox accumulation in cell nuclei is described in section 2.3.1. However, taking into account that doxorubicin fluorescence strongly depends on the environment and doxorubicin fluorescence spectra may interfere with mScarlet fluorescence the set of the experiments with the PLGA NP and hMNP co-incubation as well as the experiments with  the endocytosis inhibitors were performed using the model PLGA-Cy5 placebo NP. It is worth mentioning that the PLGA-Cy5 NP and Dox-PLGA-Cy5 NP possess very similar physicochemical parameters: they have almost the same size around 100 nm (96 and 88 nm for PLGA-Cy5 NP and Dox-PLGA-Cy5 NP, respectively) with slightly lower negative zeta potential for Dox-PLGA-Cy5 as compared to the non-loaded nanoparticles (~−14.4 mV vs ~ −30 mV, respectively).

In the first stage of the in vivo experiments, intravital microscopy was performed for the hMNP and PLGA NP separately. The biodistribution patterns of both the PLGA-Cy5 and Dox-PLGA-Cy5 NP were investigated. We added the images illustrating the Dox-PLGA-Cy5 distribution in the tumor and peritumoral vasculature and doxorubicin accumulation in the tumor tissue (Figure 11)

Moreover, this first set of IVM experiments demonstrated similar distribution patterns of the Dox-PLGA-Cy5 NP and PLGA-Cy5 NP (distribution in tumor vasculature, leakage formation) based on the PLGA-Cy5 fluorescence. 

The model PLGA-Cy5 NP were used in further experiments with different administration regimens (PLGA NP + hMNP). The use of doxorubicin-loaded nanoparticles would reduce the amount of the channels of the confocal microscope. The broad fluorescence spectra of Doxorubicin would interfere with several additional fluorophores introduced to the study: fluorescently-labeled antibodies to trace the vascular wall, immune cells. 

Of note, we are preparing other manuscripts with the detailed evaluation of the distribution of Dox-loaded PLGA NP with different drug release rates (Kovshova et al., 2023; Zhuoxuan Li et al., 2023). The main idea of the present paper was to evaluate the predictive value of the suggested theranostic pair and to choose the technique for its evaluation. 

Conclusion section is poor. Any explanations, the outstanding results should be presented. I don't clearly understand this highlights: "the model theranostic pair hMNP and Dox-PLGA NP in combination". This result is too general. I can write without the reading that good to combine diagnostic tool and drug. The novelty of the work should be mentioned in a better way. This will highlight the importance of the work.

Answer. Conclusion was revised to highlight the results of the study.

Reviewer 2 Report

The manuscript explores the combined use of HSA capsules, whose network was filled by magnetic nanoparticles, and PLGA nanoparticles loaded with doxorubicin for treating the 4T1 tumor in rats. In particular authors studied the internalization pathway of the theranostic system in vitro and its distribution within tumor cells in vivo.

The manuscript is well written, the experimental plan is accurate and the interpretation and discussion of the results are extensive. For this reason I recommend the manuscript for publication in the present version.

Some minor comments, that can be addressed during the final proofreading are reported below.

- Line 130 and line 154: the reported zeta potentials are not in accordance with the values reported in Table 1, please amend.

- Line 146: the reported hydrodynamic diameter is not in accordance with the values reported in Table 1, please amend.

- Line 169: the reference to the Figure is wrong, please correct.

- It would be useful to report in the manuscript the definitions of MOC and PCC.

- Figures 3, 4, 5, 6, 8 and 10: the font size should be increased and non-useful text should be removed to make the figures clearer.

Author Response

Some minor comments that can be addressed during the final proofreading are reported below.

- Line 130 and line 154: the reported zeta potentials are not in accordance with the values reported in Table 1, please amend.

Answer: The reported zeta potentials in the text were corrected according to the values in Table 1.

- Line 146: the reported hydrodynamic diameter is not in accordance with the values reported in Table 1, please amend.

Answer: The reported hydrodynamic diameter in the text was corrected according to the value in Table 1.

- Line 169: the reference to the Figure is wrong, please correct. 

Answer: The reference to Fig. 2 was corrected [p. 6]

- It would be useful to report in the manuscript the definitions of MOC and PCC. 

Answer: The definitions of MOC and PCC were added in the text [p. 9, section 2.3.1].

- Figures 3, 4, 5, 6, 8 and 10: the font size should be increased and non-useful text should be removed to make the figures clearer. 

Answer: The font size on the scale bars were increased in the figures. The images represented in Figures 3, 4, 5, 6, 8, and 10 represent the original images from the microscope (we are not able to increase the font size there), so we decided that this technical information shall remain. 

Some minor comments that can be addressed during the final proofreading are reported below.

- Line 130 and line 154: the reported zeta potentials are not in accordance with the values reported in Table 1, please amend.

Answer: The reported zeta potentials in the text were corrected according to the values in Table 1.

- Line 146: the reported hydrodynamic diameter is not in accordance with the values reported in Table 1, please amend.

Answer: The reported hydrodynamic diameter in the text was corrected according to the value in Table 1.

- Line 169: the reference to the Figure is wrong, please correct. 

Answer: The reference to Fig. 2 was corrected [p. 6]

- It would be useful to report in the manuscript the definitions of MOC and PCC. 

Answer: The definitions of MOC and PCC were added in the text [p. 9, section 2.3.1].

- Figures 3, 4, 5, 6, 8 and 10: the font size should be increased and non-useful text should be removed to make the figures clearer. 

Answer: The font size on the scale bars were increased in the figures. The images represented in Figures 3, 4, 5, 6, 8, and 10 represent the original images from the microscope (we are not able to increase the font size there), so we decided that this technical information shall remain. 

Round 2

Reviewer 1 Report

Thank you for the revised version. I recommend authors to think about possible features of synthesized constructions like hyperthermia and magnetic delivery experiments.